

# Unveiling aerosol-cloud interactions Part 2: Minimizing the effects of aerosol swelling and wet scavenging in ECHAM6-HAM2 for comparison to satellite data

David Neubauer[1], Matthew W. Christensen[2,3], Caroline Poulsen[2], Ulrike Lohmann[1]

[1]Institute for Atmospheric and Climate Science, ETH Zurich, Zurich, 8092, Switzerland
[2]RAL Space, STFC Rutherford Appleton Lab, Harwell, OX11 0QX, United Kingdom
[3]Atmospheric, Oceanic and Planetary Physics, University of Oxford, Oxford, OX1 3PU, United Kingdom

*Correspondence to*: David Neubauer (david.neubauer@env.ethz.ch)

**Abstract.** Aerosol-cloud interactions (ACI) are uncertain and the estimates of the ACI effective radiative forcing (ERF$_{aci}$) magnitude show a large variability. Within the Aerosol_cci project the susceptibility of cloud properties to changes in aerosol properties are derived from the high resolution AATSR dataset using the Cloud-Aerosol Pairing Algorithm (CAPA) (as described in our companion paper) and compared to susceptibilities from the global aerosol climate model ECHAM6-HAM2 and MODIS-CERES data. For ECHAM6-HAM2 the dry aerosol is analysed to mimic the effect of CAPA.
Furthermore the analysis is done for different environmental regimes.

The aerosol-liquid water path relationship in ECHAM6-HAM2 is systematically stronger than in AATSR-CAPA data and cannot be explained by an overestimation of autoconversion when using diagnostic precipitation but rather by aerosol swelling in regions where humidity is high and clouds are present. When aerosol water is removed from the analysis in ECHAM6-HAM2 the strength of the susceptibilities of liquid water path, cloud droplet number concentration and cloud
albedo as well as ERF$_{aci}$ agree much better with the ones of AATSR-CAPA or MODIS-CERES. For comparing satellite derived to model derived susceptibilities this study finds it more appropriate to use dry aerosol in the computation of model susceptibilities.

We further find that while the observed relationships of different satellite sensors (AATSR-CAPA vs. MODIS-CERES) are not always consistent for tested environmental conditions the relationships in ECHAM6-HAM2 are missing a strong
dependence on environmental conditions which is an indication that feedback processes like cloud top entrainment are missing or not well represented in the model.

Next to aerosol swelling, also wet scavenging and aerosol processing have an impact on liquid water path, cloud albedo and cloud droplet number susceptibilities. Aerosol processing leads to negative liquid water path susceptibilities to changes in aerosol index (AI) in ECHAM6-HAM2, likely due to aerosol size changes by aerosol processing. This is an indication that
AI is not necessarily a better proxy for cloud condensation nuclei than the less size dependent aerosol optical depth.





Our results indicate that for statistical analysis of aerosol-cloud interactions the unwanted effects of aerosol swelling, wet scavenging and aerosol processing need to be minimized when computing susceptibilities of cloud variables to changes in aerosol.

# 1 Introduction

Aerosol particles emitted from natural and anthropogenic sources are important for Earth's climate because of their interactions with radiation and clouds. In particular, the uncertainty of aerosol-cloud interactions is large (Boucher et al., 2013) and impairs the investigation of historical climate records and the prediction of future changes in climate. Several studies revealed differences in the response of cloud properties to changes in aerosol optical depth (AOD) in model simulations and satellite observations (e. g. Lohmann and Lesins, 2002; Quaas et al., 2009; McComiskey and Feingold,

2012; Boucher et al., 2013; Schmidt et al., 2015). These differences can be explained by the growth of aerosol particles in the humid environment surrounding clouds (Twohy et al., 2009; Boucher and Quaas, 2012), misclassification of partly cloudy satellite pixels as cloud free (cloud contamination), brightening of aerosol particles by sunlight reflected at the edge of clouds (3D-effects; Varnái and Marshak, 2009), processing of aerosol particles in clouds by nucleation or impact scavenging, subsequent growth by heterogeneous chemistry and re-evaporation, wet scavenging of aerosol particles in

particular in areas of strong precipitation (Grandey et al., 2014; Gryspeerdt et al., 2015), by stability/humidity changes due to absorbing aerosol above/near clouds, structural uncertainties due to differences in the analysis/observational scale and the process scale (McComiskey and Feingold, 2012), or co-variation of aerosol and cloud properties with meteorology (Chen et al., 2014; Andersen et al., 2017). Andersen et al. (2016) showed that cloud droplet size sensitivity to aerosol loading depends on the magnitude of the aerosol loading and that the magnitude of greatest sensitivity is larger for larger total columnar water

vapour (with a possible explanation being aerosol swelling). Quaas et al. (2010) identified the swelling of aerosols (Zhao et al., 2017) as the most likely explanation of the larger cloud cover susceptibility (to AOD) in observations than in models. Gryspeerdt et al. (2014) showed that the cloud top height susceptibility is not a direct response to aerosol changes but mediated by changes in cloud cover (which as the study by Quaas et al., 2010 showed is likely due to covariation of relative humidity). To circumvent the covariation of relative humidity in the cloud cover susceptibility, Gryspeerdt et al. (2016) use

the cloud droplet number susceptibility to mediate the cloud cover susceptibility. Thus, cloud cover can only change through a change in cloud droplet number concentration. The mediated cloud cover susceptibilities are much smaller than the 'direct' cloud cover susceptibility, hinting at the large influence of other factors like humidity. Bender et al. (2016) used a different approach for analysing albedo-cloud cover histograms. Because of the correlation of cloud cover and AOD they subtract for each cloud cover bin the mean AOD to obtain the correlation of AOD anomalies to the albedo-cloud cover histograms. After

the subtraction they find indications that absorbing aerosol influences the cloud albedo in Namibian and Canarian Stratocumulus regions. Boucher and Quaas (2012) and Grandey et al. (2014) used dry AOD to remove the effect of humidity on the susceptibility of the precipitation rate to changes in AOD. Koren et al. (2013) on the other hand showed with basic



hygroscopic growth and radiative transfer calculations that aerosol swelling alone cannot explain the large difference in AOD in polluted and clean conditions. The algorithm applied for the MODIS AOD product that they used filters pixels within 1 km of detectable clouds and 25% of the brightest pixels are rejected within each $10 \times 10$ km aerosol retrieval box. This should significantly reduce the effect of hygroscopic growth and is similar to the minimum distance applied in the Cloud-Aerosol Pairing Algorithm (CAPA) for the AATSR and MODIS products in our study.

The liquid water path (LWP) response to AOD changes also shows a difference between model simulations and satellite observations, such that it is in general larger in model simulations than in satellite observations (Quaas et al, 2009). Although this difference can be explained by similar influences as for the cloud cover susceptibility, it also depends on the ratio (autoconversion rate / autoconversion rate + accretion rate) of the processes contributing to precipitation formation in global model simulations (Posselt and Lohmann, 2009; Quaas et al, 2009; Gettelman et al., 2015; Sant et al., 2015). We investigate the importance of how precipitation formation is simulated with a prognostic precipitation scheme using prognostic variables for snow, rain and drizzle (Sant et al., 2015). Similar to the cloud cover susceptibility, the LWP susceptibility (to aerosol changes) is affected by humidity. In the companion paper Christensen et al. (2017) the effects of aerosol swelling, cloud contamination and 3D-effects are reduced by using a minimum distance between aerosol and cloud observations after screening for contaminated aerosol in the vicinity of clouds. In a global model with its coarse resolution a similar approach is not feasible, therefore we evaluate the susceptibilities with respect to dry aerosol, which is similar to CAPA in Christensen et al. (2017). By removing the overshadowing effect of aerosol swelling in the global aerosol-climate model ECHAM6-HAM2 we can also identify other processes influencing the studied susceptibilities.

For studying aerosol-cloud interactions in observational data a proxy for cloud condensation nuclei (CCN) is necessary. Liu and Li (2014) show based on surface measurements show that aerosol index (AI) is a better proxy for CCN than AOD and that in situ scattering AI at the surface (i.e. not vertically integrated) has the highest correlation to CCN at the surface. Stier (2016) has shown using model simulations that vertically resolved measurements of aerosol radiative properties (i.e. as a function of altitude) would be necessary to obtain a good CCN proxy for most of the globe. In the absence of vertical information AI is considered better as a CCN proxy than AOD due to the higher weight of smaller aerosols at larger optical depths (Nakajima et al., 2001). The simulations by Stier (2016) and surface measurements do not account for aerosol processing in clouds, which can affect the suitability of aerosol properties as CCN proxy. Shinozuka et al. (2015) propose to use the in situ dry extinction coefficient and Ångström exponent to parameterize CCN, which accounts for ambient relative humidity, vertical information and aerosol size. Interestingly in the parameterization of Shinozuka et al. (2015) the CCNs do not increase linear with the dry extinction coefficient which is an indication of growth processes like condensation, coagulation or in-cloud aerosol processing. Aerosol particles can activate as CCN, collide and coalesce with cloud droplets and atmospheric gases can be taken up by cloud droplets und undergo chemical reactions in the aqueous phase. Aerosol particles release by evaporation of cloud droplets or rain drops are larger than before the processing in the clouds. We compare simulations with and without aerosol processing in clouds to obtain an indication of how aerosol processing affects the suitability of different aerosol properties as proxies for CCN.





In section (2) the methodology is outlined and satellite products and model experiments are described in section (3). The results are presented in section (4) and summarized in section (5) where also conclusions are drawn.

## 2 Methodology

For a statistical analysis of aerosol-cloud interactions from satellite data, the data from aerosol and cloud retrievals need to be paired. The Cloud-Aerosol Pairing Algorithm (CAPA) used here for the satellite data is described in subsection 2.1. In a model on the other hand, the model parameterizations use the aerosol in a grid box to compute cloud microphysical processes, so the aerosol and cloud data in a grid box match each other all the time due to the model parameterizations and no further association is necessary. The computation of susceptibilities for the paired aerosol and cloud data from satellite products and the model data is described in subsection 2.2. As a proxy for CCN, the AI is used. AI is computed by multiplying AOD by the Ångström exponent (AE). For ECHAM6-HAM2 and the Aerosol_cci products we compute the Ångström exponent from AOD at 550 nm and 865 nm (see subsection 2.3). For the Cloud_cci AATSR products the effective cloud droplet number concentration (CDNC) is derived. By combining Eqs. (7) and (10) from Bennartz (2007) and assuming a cloud fraction=1, $N_d$ can be written as:

$$CDNC = \frac{1}{2\pi \cdot k} \cdot \sqrt{\frac{5 \cdot c_W \cdot COD}{Q_{ext} \cdot \rho_{H_2O} \cdot R_e^5}} = \gamma \cdot COD^{0.5} \cdot R_e^{-2.5} \qquad (1)$$

with

$$\gamma = \frac{1}{2\pi \cdot k} \cdot \sqrt{\frac{5 \cdot c_W}{Q_{ext} \cdot \rho_{H_2O}}} = 1.37 \cdot 10^{-5} m^{-0.5} \qquad (2)$$

COD is cloud optical depth and $R_e$ is the cloud top droplet effective radius. Further variables are as defined in Bennartz (2007). Bennartz (2007) discusses the contribution of the variables in Eq. (1) to the uncertainty in CDNC and find that the three input parameters $Q_{ext}$, $c_W$ and $k$, summarized in γ, together only account for about 15% of the total variance in CDNC. Therefore, in the literature often a constant value for γ is used. The value for γ in Eq. (2) is from Quaas et al. (2006) derived from constants in Brenguier et al. (2000). Eq. (1,2) assume cloud adiabatic growth. Zeng et al. (2014) compare CDNC computed from the passive sensor MODIS to CDNC from CALISPO depolarization measurements which do not rely on the adiabatic assumption (using $r_e$ from MODIS/PARASOL). In regions where clouds grow adiabatic (like stratocumulus regions) the agreement between the two methods is reasonable.

The cloud albedo ($\alpha$) of ECHAM6-HAM2 is computed from shortwave fluxes at the top of the atmosphere:

$$\alpha = \frac{F_{cld}^u}{F_{cld}^d} \qquad (3)$$

Where $F_{cld}^u$ and $F_{cld}^d$ denote top of the atmosphere up- and downwelling shortwave fluxes in the cloudy part of the model grid column. As $\alpha$ is calculated from shortwave fluxes the $\alpha$ susceptibility can only be calculated during the day and therefore is computed from fewer aerosol-cloud data pairs than the other susceptibilities.



## 2.1 Cloud-Aerosol Pairing Algorithm (CAPA)

CAPA applied to pair aerosol and cloud pixels is described in detail in the companion paper Christensen et al. (2017). By pairing high resolution retrievals of aerosol and cloud properties CAPA aims at minimizing data aggregation effects at coarser resolution (McComiskey and Feingold, 2012) and provides sufficient data pairs for significant susceptibilities. To reduce cloud contamination, 3D-effects and aerosol swelling, a minimum distance of 15 km is required between the aerosol and cloud pixels.

## 2.2 Susceptibility computation

Susceptibilities ($ACI_y$) are computed at the highest spatial resolution available by linear regression over all aerosol-cloud data pairs of a season following Feingold et al. (2003):

$$ACI_y = \frac{d \ln y}{d \ln x} = \frac{\sum_{i=1}^{n}(\ln x_i - \overline{\ln x})(\ln y_i - \overline{\ln y})}{\sum_{i=1}^{n}(\ln x_i - \overline{\ln x})^2} \tag{4}$$

$$\sigma_{ACI_y} = \sqrt{\frac{\sum_{i=1}^{n}(\ln y_i - \overline{\ln y})^2 / \sum_{i=1}^{n}(\ln x_i - \overline{\ln x})^2 - (ACI_y)^2}{n-2}} \tag{5}$$

where $y$ is a cloud property like LWP and $x$ is the aerosol property like AI. The natural logarithm of $x$ and $y$ is used to make the susceptibilities $ACI_y$ independent of the units used for $x$ and $y$. We require a minimum number of aerosol-cloud data pairs $n \geq 100$ for the computation of the linear regression (for the 12/18 years of analysed model data; note that the high resolution satellite data using CAPA provides many more data pairs). Averages over larger areas and/or longer time spans use the weighted mean method by Grandey and Stier (2010). As weights for each grid point the inverse of the standard deviation of the linear regression given in Eq. (5) : $\left(\sigma_{ACI_y}\right)^{-2}$ is used in Grandey and Stier (2010), which makes the weights approximately proportional to the number of aerosol-cloud data pairs $n$ used in the linear regression. As this sigma error weighting could lead to a bias towards regions and seasons with low one-sigma error, we use the number of aerosol-cloud data pairs $n$ as weights instead:

$$\overline{ACI_y} = \frac{\sum_{k=1}^{m} ACI_{y,k} \cdot n_k}{\sum_{k=1}^{m} n_k} \tag{6}$$

$$\sigma_{\overline{ACI_y}} = \sqrt{\frac{\sum_{k=1}^{m}(ACI_{y,k} - \overline{ACI_y})^2}{m-1}} \tag{7}$$

Where $k = 1, \cdots, m$ is the index over all susceptibilities $ACI_{y,k}$ computed at high resolution (e.g. 1° spatial resolution) in a larger region consisting of $m$ high resolution grid areas (see Fig. 1 in Grandey and Stier, 2010). Because of the known issues of satellite observations at high zenith angles and over bright surfaces (see e.g. Zygmuntowska et al., 2012) high latitudes (> 60°N and > 60°S) have been excluded in this analysis. The analysis is done for eight different environmental regimes defined by the amount of precipitation, humidity in the free troposphere and stability of the lower troposphere and calculated separately for land and ocean. Moist conditions stand for free tropospheric relative humidity > 40% and dry for < 40%,





stable conditions stand for lower tropospheric stability > 17 K and unstable for <17 K. The lower tropospheric stability (LTS) is computed as the difference in potential temperature at 700 hPa and the surface:

$$LTS = \theta_{700hPa} - \theta_{Surface} \tag{8}$$

The free tropospheric humidity ($RH_{FT}$) is defined as the average relative humidity between 850 hPa and and 700 hPa:

$$RH_{FT} = \sum_{850hPa}^{700hPa} RH \, / n_l \tag{9}$$

where $n_l$ is the number of levels between 850hPa and 700 hPa. Raining and non-raining scenes are either differentiated by model precipitation (smaller or larger 0.5 mm / day), by the CloudSat precipitation flag or by using $R_e$ of 14 μm as a proxy for precipitation (Rosenfeld et al., 2014), where $R_e$ > 14 μm is a proxy for raining scenes and $R_e \leq$ 14 μm for non-raining scenes. Fig. 1 shows the frequency of occurrence of all environmental regimes.

Our analysis uses the pixel-scale (1-km spatial resolution) Level 2 Aerosol and Cloud_cci AATSR products. Only data points are analysed where (fully overcast) cloud and aerosol pixels can be paired using CAPA. The AATSR cloud properties therefore represent in-cloud properties. The ECHAM6-HAM2 cloud properties are divided by to the low liquid cloud cover (cloud top pressures > 500 hPa and cloud top temperatures > 273.15 K) to obtain in-cloud values also for the global model data. The computation of mean susceptibilities in Eq. (6) uses the number of aerosol-cloud data pairs $n$ which is a subsample

of the number of cloudy pixels. The susceptibilities computed by Eq. (6) represent therefore grid-mean values (in-cloud $ACI$ multiplied by $n$).

Susceptibilities are computed for each grid area for each season using all available years (e.g. all summer seasons during 1995-2012 for the model data, 2002-2012 for AATSR data and 2006-2010 for MODIS data). Annual mean susceptibilities are computed as a weighted mean from the seasonal susceptibilities.

To assess the importance of relative humidity on aerosol-cloud susceptibilities multiple linear regression could be used in principle. Due to the non-linear dependence of AOD and cloud properties on relative humidity, the ambient relative humidity would need to be observed with high precision at high-resolution (horizontal and vertical). As such high-resolution satellite observations of humidity are not available we use therefore CAPA for AATSR products and remove aerosol water from AOD and AI in ECHAM6-HAM2 data.

**2.3 Aerosol index and dry aerosol**

The AI is computed as the product of AOD and the Ångström exponent ($ANG$; Angstrom, 1964):

$$AI = AOD_{550} \times ANG \tag{10}$$

The Ångström exponent is computed from AOD at 550nm and 865nm:

$$ANG = -(\log AOD_{550}/AOD_{865})/(\log 550nm/865nm) \tag{11}$$

For the dry aerosol properties the water taken up by the aerosol particles is removed:

$$AODdry = AOD - AOD_{aerosol\ water} \tag{12}$$

$$AIdry = AODdry_{550} \times (\log AODdry_{865}/AODdry_{550})/(\log 550nm/865nm) \tag{13}$$



This is done only diagnostically, in the simulations the normal AOD including aerosol water is used.

**2.3 Effective radiative forcing**

The effective radiative forcing due to aerosol-cloud interactions (ERF$_{aci}$) is estimated from the top of the atmosphere clear-sky ($\alpha_{clr}$) and $\alpha$ following Chen et al. (2014) and Christensen et al. (2017):

$$ERF_{aci} = \left( \overline{LCC_m} \left[ \frac{d\alpha_{clr}}{d \ln AI} - \frac{d\alpha}{d \ln AI} \right] - \overline{\alpha_{clr} - \alpha} \frac{dLCC}{d \ln AI} \right) \Delta a F_d \qquad (14)$$

Where $\overline{LCC_m}$ is the annual mean low liquid cloud cover, $\overline{\alpha_{clr} - \alpha}$ is the annual mean shortwave clear-sky minus cloud albedo, $\Delta a = \ln \frac{AOD}{AOD - AOD_{anth}}$ represents the anthropogenic aerosol increase ($AOD_{anth}$ is anthropogenic AOD), which is taken from Bellouin et al. (2013) for AATSR and from reference model simulations (Neubauer et al, 2014) for ECHAM6-HAM2. Note that $\Delta a$ based upon AOD can be used because in general $\frac{d\alpha}{d \ln AOD} \sim \frac{d\alpha}{d \ln AI}$. F$_d$ is the mean incoming solar radiation. The

methodology of Quaas et al. (2008), separates the total anthropogenic aerosol forcing into the increase in CDNC and hence $\alpha$ at constant LWP (first indirect aerosol effect; Twomey, 1974) and a remainder that encompasses the changes in cloud cover and LWP (second indirect effect; Albrecht, 1989) and possible other processes and artefacts. In contrast, this methodology separates the total anthropogenic aerosol forcing into the change in cloud cover (called extrinsic forcing) and the changes in $\alpha$ where LWP is allowed to change (called intrinsic forcing).

**3 Satellite products and model experiment description**

**3.1 Satellite products**

Data for the environmental conditions is taken for both satellite datasets (AATSR and MODIS) from the European Center for Medium range Weather Forecast-AUXiliary analysis (ECMWF-AUX) product.

**3.1.1 AATSR Aerosol_cci/Cloud_cci products**

The susceptibilities for the Advanced Along-Track Scanning Radiometer (AATSR) data have been computed with CAPA described in Christensen et al. (2017) from the ESA Aerosol_cci L2 aerosol products, ORAC V4.01 which are available at 10x10 km horizontal resolution (Popp et al., 2016) and the ESA Cloud_cci L2 cloud products, ORAC V4.01 which are available at 1x1 km horizontal resolution (Hollmann et al., 2013). The aerosol and cloud products use a similar optimal estimation algorithm (Thomas et al., 2009; Poulsen et al., 2012) and efforts are made within the Aerosol_cci and Cloud_cci

projects to ensure that consistent cloud masking is used in the products. AOD at 550 and 865 nm, R$_e$, cloud LWP, cloud ice water path, cloud optical thickness, cloud top pressure, and cloud top temperature are taken directly from Aerosol_cci and Cloud_cci products and from this additional variables were derived as described in section 2. Ten years of data from 2002 to 2012 are used for computing susceptibilities and forcing estimates.



### 3.1.2 MODIS/CERES/CloudSat products

The A-train satellite products are the same as described in Christensen et al. (2016). The data include CloudSat radar data, CERES (Clouds and the Earth's Radiant Energy System) radiative fluxes and Moderate Resolution Imaging Spectroradiometer (MODIS) level 2 (MYD06) cloud and MODIS (MYD08) aerosol products. The methodology is following Chen et al. (2014). All sensors were matched to the nearest CloudSat footprint. The CloudSat precipitation flag is used to identify raining scenes.

Aerosol data are taken from the gridded MODIS (MYD08) atmospheric product (1° x 1°) which is based on the MYD04 aerosol product at 10 x 10 km. For the MYD04 aerosol product only those retrieved pixels at 1 x 1 km are used that are considered cloud-free (elimination of spatially inhomogeneous 3x3 pixel groups and of the darkest and brightest 25% of pixels within 10 km x 10 km boxes; Remer et al., 2005) in averaging to 10 x 10 km resolution to limit cloud contamination. Data for 2006-2010 was used for comparability with Chen et al. (2014). For the sake of brevity these products are referred to as MODIS-CERES (note that the MODIS-CERES forcing data are taken from Chen et al., 2014).

### 3.2 ECHAM6-HAM2 experiments

### 3.2.1 Model description

ECHAM-HAMMOZ is a global aerosol-chemistry climate model of which in this study only the global aerosol-climate model part is used. Two versions of ECHAM-HAM are used because they have different options to treat aerosol-cloud interactions. ECHAM6.1-HAM2.2 (Neubauer et al., 2014), for the sake of brevity referred to as ECHAM6-HAM2, consists of the general circulation model ECHAM6 (Stevens et al., 2013) coupled to the aerosol module HAM2 (Zhang et al., 2012), which includes a size-dependent in-cloud scavenging parameterization (Croft et al., 2010). ECHAM5.5-HAM, for the sake of brevity referred to as ECHAM5-HAM, consists of the general circulation model ECHAM5 (Roeckner et al., 2003) coupled to the aerosol module HAM (Stier et al., 2005). Some of the model components of ECHAM6-HAM2 and ECHAM5-HAM are similar although in ECHAM6-HAM2 several software errors have been fixed. Both model versions use a two-moment cloud microphysics scheme which solves prognostic equations for both mass mixing ratios and number concentrations of cloud liquid water and cloud ice (Lohmann et al., 2007; Lohmann and Hoose, 2009). The Lin and Leaitch (1997) aerosol activation scheme and the Khairoutdinov and Kogan (2000) autoconversion scheme are used in both model versions as well. Also the Tiedtke (1989) convection scheme with modifications by Nordeng (1994) for deep convection is used in both model versions. Futhermore in both, ECHAM6-HAM2 and ECHAM5-HAM, aerosol effects on convective clouds are not included, but there is a dependence of cloud droplets detrained from convective clouds on aerosol. In order to facilitate the comparability of the numerical experiments of both model versions all simulations were performed with the same resolution, T63 (1.9° × 1.9°) horizontal spectral resolution using 31 vertical levels (L31).

ECHAM6-HAM2 and ECHAM5-HAM use a 1.5 order turbulence closure scheme with a simplified prognostic equation for turbulent kinetic energy (TKE) (Brinkop and Roeckner, 1995) to compute vertical diffusion (mixing) in the boundary layer.





In the ECHAM6-HAM2 simulation with aerosol processing in stratiform clouds, the scheme from Hoose et al. (2008a, b) is applied in order to extend the seven aerosol modes of HAM2 through an explicit representation of aerosol particles in cloud droplets and ice crystals in stratiform clouds. The in-cloud aerosol is represented by five tracers for sulphate, black carbon, organic carbon, sea salt and mineral dust for each, cloud droplets and ice crystals (see details in Neubauer et al., 2014).

Aerosol mass transfers to and from these in-cloud aerosol tracers by nucleation and impact scavenging, freezing and evaporation of cloud droplets, and melting and sublimation of ice crystals are computed explicitly. Aerosol particles from evaporating/sublimating clouds and precipitation are released to the modes that correspond to their size.

In the ECHAM5-HAM simulation with prognostic precipitation, the prognostic precipitation scheme by Sant et al. (2015), which builds on work by Posselt and Lohmann (2008) and Sant et al. (2013), is applied which uses in addition to the

standard cloud liquid water and cloud ice classes also rain, drizzle and snow. For all five water classes (three liquid, two solid) prognostic equations for both mass mixing ratios and number concentrations are solved.

### 3.2.2 Experiment description

The experiment setup follows the guidelines of the AeroCom aerosol-climate model intercomparison initiative (http://aerocom.met.no/) Phase III intercomparison on assessing the aerosol indirect effect

(https://wiki.met.no/aerocom/indirect). The length of the simulations was 18 years (1995–2012) after 3 months of spin-up to get enough aerosol-cloud data pairs for significant susceptibilities. Present-day (year 2000) greenhouse gas concentrations were used in all simulations. Each experiment uses present-day (year 2000) aerosol emissions from the AeroCom Phase II data set (ACCMIP by Angelika Heil, Martin Schultz and colleagues; see http://aerocom.met.no/emissions.html; Lamarque et al., 2010). The simulations were conducted with sea surface temperatures and sea ice cover fixed to observed values (AMIP

simulations). In all simulations winds and temperatures were nudged towards ERA-Interim (Dee et al., 2011) reanalysis. 3-hourly instantaneous output is used. The effect of using 3-hourly output and not only output at the time of the satellite overpass is discussed in Appendix A. For cloud top values (e.g. CDNC) the maximum-random overlap assumption is used to compute 2D-fields.

To focus only on warm, liquid clouds in the analysis, model cloud top pressure and temperature are used to identify low

liquid clouds as those with cloud top pressures greater than 500 hPa and cloud top temperatures exceeding 273.15 K. The model variables are used for the sampling and environmental regime discrimination for the model data. Minimum and maximum values for aerosol and cloud properties are applied to mimic the sensitivity of the satellite retrievals and remove unrealistically large values that could influence the linear regression (Table 1). The same conditions (cloud type and environmental conditions) on the selection criteria are used for the satellite analysis (environmental data was taken from the

ECMWF-AUX product).

Four experiments were conducted: a reference simulation with ECHAM5-HAM (E5_Ref), a reference simulation with ECHAM6-HAM2 (E6_Ref), a simulation with ECHAM5-HAM and the prognostic precipitation scheme (E5_Prog) and a simulation with ECHAM6-HAM2 and the aerosol processing scheme (E6_AProc). The E5_Ref and E5_Prog simulations



were run for 12 years (2000-2011) as some input files for this older ECHAM-HAM version were not available for the years 1995-1999 and 2012.

## 4 Results

### 4.1 Satellite and model susceptibilities

### 4.1.1 Impact of humidity, wet scavenging and aerosol processing

In Fig. 2a the annual mean susceptibility of the LWP to changes in AI during 1995-2012 between 60°N and 60°S is shown. The LWP susceptibility is positive almost everywhere (i.e. an increase in AI leads to an increase in LWP and a decrease in AI leads to a decrease in LWP) and the LWP susceptibility is relatively large in many areas. In Fig. 2b the same is shown as in Fig. 2a, only AIdry (without aerosol water) was used in the computation of the susceptibility. The effect of removing the

water uptake by aerosol is immense. Large parts of the globe now show a negative LWP susceptibility (an increase in AI leads to a decrease in LWP and a decrease in AI leads to an increase in LWP) when AIdry is used. In areas where the LWP susceptibility is positive its magnitude is smaller than in Fig. 2a. AI and AIdry are used as a proxy for CCN in the study of aerosol-cloud relationships. AIdry is better suited due to the water uptake of aerosol particles in the humid environment close to clouds that affects AI and masks the true effects of the aerosol particles. This co-variation with relative humidity is

accounted for when using AIdry. The comparison between Fig. 2a and Fig. 2b shows that the applicability of AI as a CCN proxy is limited by aerosol swelling. Aerosol water uptake in ECHAM6-HAM2 is large, 66% of the total aerosol mass burden is composed of aerosol water. This is well within the range for water uptake of an AeroCom intercomparison study (Textor et al., 2006) where the multi model mean and diversity aerosol water mass burden fraction was 48% ± 42% (excluding one outlier).

To further remove the effects of covarying variables, in Fig. 2c the LWP susceptibility to changes in AIdry is shown only for non-raining scenes. This minimizes the effect of wet scavenging of aerosol particles by precipitation but cannot fully remove it (Gryspeerdt et al., 2015). Clouds with higher LWP are more likely to remove aerosol particles by wet scavenging leading to a negative LWP susceptibility in particular in regions where heavy precipitation occurs frequently. In Fig. 2c the LWP susceptibility is positive almost everywhere except in regions where deep convection and moderate and heavy precipitation

are frequent, so the negative LWP susceptibilities in Fig. 2b seem to be due to wet scavenging. Moderate and heavy precipitation originates predominantly from convective clouds in ECHAM6-HAM2 whereas light precipitation comes mainly from stratiform clouds. In Fig. 2c the LWP susceptibility of precipitating convective clouds is therefore still largely masked by wet scavenging. In Fig. 2a the effect of wet scavenging is not as easily identifiable as in Fig. 2b as the effect of aerosol swelling is overshadowing other factors that influence the statistical relationship of LWP and aerosol like wet

scavenging. In Fig. 2d the same is shown as in Fig. 2c but using a $R_e \leq 14$ μm to identify non-raining scenes. This leads to are more areas where the LWP susceptibility is negative than in Fig. 2c though less than in Fig. 2b. Further studies will be



necessary to assess the usability of $R_e$ as a proxy for precipitation state. In ECHAM6-HAM2 a $R_e$ of 14 μm shows deficiencies as a proxy for precipitation state when analysing aerosol-cloud relationships (a combination of $R_e$ (> 14 μm for rain) and LWP (> 100 g/m$^2$ for rain) as a proxy for precipitation state as indicated in Stephens et al., (2008) gives similar results to the $R_e$ criterion alone in ECHAM6-HAM2).

Fig. 2e shows the same as Fig. 2c but for the simulation with processing of aerosol in stratiform clouds. The LWP susceptibility is negative almost everywhere in Fig. 2e although only non-raining scenes are shown, i.e. the effect of wet scavenging should be minimal. The negative LWP susceptibilities can be explained by the growth of aerosol particles in cloud droplets (by collisions of the cloud droplets with interstitial aerosol particles and heterogeneous chemistry) and release of the larger aerosol particles when the cloud droplets evaporate (as AIdry decreases for larger particles). The larger the

LWP (or cloud lifetime), the more aerosol may be processed and grow in size in the cloud and therefore lead to negative LWP susceptibilities to changes in AIdry. A further indication that the negative LWP susceptibility in Fig. 2e is due the growth of aerosol particles by aerosol processing is that the LWP susceptibility to changes in AODdry is positive in most regions (see Fig. 2f) even with aerosol processing. AODdry is less sensitive to aerosol size than AIdry so the LWP susceptibility shown in Fig. 2f should represent the direct relationship of aerosol and LWP and not the effect of aerosol

processing (for comparison the LWP susceptibility to changes in AODdry is shown in Fig. 2g). This would be an indication that AODdry is a better proxy for CCN than AIdry. It should be noted here that ECHAM6-HAM2 overestimates the lifetime of sea salt particles when aerosol processing is used (Hoose et al., 2008a) and it uses a modal approach to simulate aerosol size and this may be too coarse to well capture the size changes by aerosol processing. Because of these limitations of ECHAM6-HAM2 we use both AI/AIdry as a proxies for CCN in this study. Further research for example using a bin

representation of aerosol size could give further insight of the effect of aerosol processing on aerosol-cloud relationships and the usability of AODdry and AIdry as proxies for CCN.

In Fig. 2a-e the regions over the oceans, where typically shallow convective clouds are present, show a particularly strong LWP susceptibility (positive or negative). In Fig. 2 in-cloud susceptibilities are shown and in these regions of high LWP susceptibility clouds are not frequent (see Fig. 1) so these regions do not contribute much to global or regional mean

susceptibilities.

### 4.1.2 Impact of environmental regimes

The response of CDNC to changes in AI (dlnCDNC/dlnAI) over the global oceans is shown in Fig. 3. For ECHAM6-HAM2 this CDNC susceptibility varies only little between moist or dry free tropospheric conditions and a stable or unstable lower troposphere with regime average values around ~0.3 for AI. For AIdry the CDNC susceptibility is generally smaller, up to

50% less depending on the regime. The differences between non-raining and raining scenes are small and in general the CDNC susceptibility is smaller in the raining scenes than in the non-raining scenes which is an indication of wet scavenging affecting aerosol concentrations in the raining scenes.



The response of LWP to changes in AI (dlnLWP/dlnAI) shown in Fig. 4, reveals again larger susceptibilities and lower variability in susceptibilities between environmental regimes in ECHAM6-HAM2 than in satellite observations when AI is used. When AIdry is used instead the magnitude of the LWP susceptibility is close to that of AATSR-CAPA and MODIS-CERES but the variability between environmental regimes is still low in ECHAM6-HAM2. A reason for the insensitivity to

environmental regimes in ECHAM6-HAM2 could be that cloud-top entrainment is not well represented in the model. With the TKE scheme used in ECHAM for boundary layer mixing it should in principle be possible to compute cloud-top entrainment when a fine vertical resolution is used. At the coarse vertical resolution of a GCM numerical artefacts like numerical entrainment (Lenderink and Holtslag, 2000) occur and the cloud top cooling that drives the turbulence in the boundary layer cannot be computed accurately (Stevens et al., 1999). A better representation of cloud-top entrainment could

act as a buffering mechanism (Stevens and Feingold, 2009) and reduce the LWP susceptibility in ECHAM6-HAM2 in unstable and/or dry regimes. Also using a prognostic precipitation scheme does not increase the sensitivity to environmental regimes in ECHAM5-HAM (not shown).

In the AATSR-CAPA dataset the variability is similar to ECHAM6-HAM2. In most regimes the LWP susceptibility is larger in the non-raining scenes than in the raining scenes. In the non-raining scenes of the MODIS-CERES data the LWP

susceptibility is negative which could be an indication of cloud-top entrainment. Chen et al. (2014) found negative LWP susceptibilities in all environmental regimes for non-raining scenes from MODIS-CERES as shown in Fig. 4. They attribute this to entrainment of dry and warm air from the free troposphere into the boundary layer due to decreased cloud droplet sedimentation of smaller cloud droplets at higher AI. The entrainment is stronger if the free troposphere is drier and/or the lower troposphere is more unstable. Although AATSR-CAPA and MODIS-CERES observed similar scenes, this effect of

entrainment seems not to appear in the non-raining scenes in the AATSR-CAPA data. A reason could be the different sampling between AATSR-CAPA and MODIS-CERES where AATSR has a longer time series but MODIS has a wider swath. Other differences could be in the retrieval scheme used to obtain cloud and the aerosol properties - ORAC which uses an optimal estimation method to acquire radiative consistency in the retrieval using all of the channels simultaneously compared to MODIS which uses discrete channel selection to retrieve aerosol and cloud properties (King et al., 1998)

separately. The aerosol retrieval has been validated and evaluated within ESA's Aerosol_cci project and a comparable quality of the AATSR and MODIS aerosol retrievals over ocean has been found (Popp et al., 2016). Another reason could be that a $R_e$ of 14 µm is not a good proxy for precipitation state of AATSR data (see subsection 4.1.1).

Next to changes in cloud microphysical parameters (CDNC, LWP) it is also interesting to investigate the impact of changes in aerosol on a cloud macrophysical parameter like $\alpha$ which is closely related to the effective radiative forcing. The

uncertainties in $\alpha$ are better known than for other cloud parameters as less assumptions are made in its computation from retrieved cloud properties (Feingold et al., 2016). The susceptibility of $\alpha$ is weaker than the CDNC or LWP susceptibility to changes in AI (AIdry) in both the model and the satellite data (Fig. 5). As for the two other susceptibilities, also for the $\alpha$ susceptibility the magnitude of the susceptibility is weaker when aerosol water is eliminated from the analysis (AIdry). Also the dependence on environmental regime is weak in ECHAM6-HAM2 also for the $\alpha$ susceptibility, except for the





susceptibility of $\alpha$ to changes in AIdry which is stronger for the unstable than the stable regimes (see Fig. 5). In the raining scenes the $\alpha$ susceptibility is weaker than in the non-raining scenes or even negative for the moist/stable and dry/unstable regimes ($R_e$ increases in these regimes in the raining scenes – not shown). This is another indication that wet scavenging in the raining scenes affects AI and AIdry and that the $\alpha$ susceptibility in the raining scenes not only represents the effect of
aerosol on clouds but also the effect (mediated by precipitation) of clouds on aerosol.

### 4.1.3 Impact of prognostic precipitation scheme

For the evaluation of the impact of a prognostic precipitation scheme on aerosol susceptibilities we use the prognostic precipitation scheme developed by Sant et al. (2013), which has recently been implemented in ECHAM5-HAM (Sant et al., 2015) and solves prognostic equations for rain, drizzle and snow. Compared to conventional prognostic precipitation
schemes, the additional drizzle class allows to better represent the drop size distribution and the drizzling conditions that often occur in marine stratocumulus clouds. Previous studies found a shift of precipitation formation from autoconversion to accretion when using a prognostic instead of a diagnostic precipitation scheme, in better agreement with observations (Posselt and Lohmann, 2008; Gettelman and Morrison, 2015). The change to a prognostic precipitation scheme or an autoconversion scheme that depends less on the CDNC results in a smaller effective radiative forcing due to aerosol-
radiation and aerosol-cloud interactions (ERF$_{ari+aci}$) (Menon et al., 2002; Rotstayn and Liu, 2005, Penner et al., 2006; Posselt and Lohmann, 2009; Gettelman et al, 2015) as accretion is independent of the CDNC. Sant et al. (2015) also find a strong shift of precipitation formation from autoconversion to accretion and a smaller increase of the cloud LWP due to anthropogenic aerosol with their prognostic precipitation scheme. ERF$_{ari+aci}$ however was stronger in their simulation with the prognostic precipitation scheme than with the diagnostic precipitation scheme. In agreement with this increase in
ERF$_{ari+aci}$ we also find stronger susceptibilities in the E5_Prog simulations compared to E5_Ref as shown in Fig. 6 for the LWP susceptibility (dlnLWP/dlnAI) for E5_Prog, E5_Ref and E6_Ref. The LWP susceptibility in E5_Prog is almost twice as large as in E5_Ref for many environmental regimes for both non-raining and raining scenes. E5_Prog and E5_Ref only differ by the precipitation scheme, in particular the autoconversion parameterization, but the model's tuning parameters are the same. A similar increase occurs for other susceptibilities (not shown). There are two reasons for this. First the LWP in
stratocumulus regions is higher in E5_Prog than in E5_Ref (Fig. 7a) because of the change of rain (E5_Ref) to drizzle (E5_Prog) in these regions. The increased LWP in E5_Prog (and the increased variability in LWP), seem to increase the (present day) LWP susceptibility in these regions. This is in contrast to the smaller increase of LWP due to anthropogenic aerosol reported in Sant et al. (2015), who computed this increase from simulations with present day versus pre-industrial aerosol. Carslaw et al. (2013) and in a recent study Ghan et al. (2016) found that present day variability is a poor proxy for
the change due to anthropogenic aerosol. Our results corroborate these findings as the difference between the prognostic and the diagnostic precipitation scheme leads to a weaker LWP response to anthropogenic aerosol (Sant et al., 2015) but a stronger LWP response determined by present day variability (Fig. 6). The other reason for the stronger response of LWP to AI is that AI is larger in E5_Prog than in E5_Ref over the oceans. This leads to a general increase of the susceptibilities.

Because AOD is more closely related to the aerosol mass whereas AI also takes into account the aerosol size, it is instructive to compare AOD and AI in E5_Prog and E5_Ref as it gives an indication whether smaller or larger particles are removed more efficiently by the different precipitation schemes. The AOD is smaller in E5_Prog than in E5_Ref whereas AI is larger over the oceans in E5_Prog than in E5_Ref (in the global mean AI is similar in E5_Prog and E5_Ref). The prognostic

precipitation scheme therefore seems to remove more efficiently larger aerosol particles than the diagnostic precipitation scheme.

These differences in LWP and AI between the simulations have a strong impact on the computed susceptibilities. Global observations with a low uncertainty would be necessary to constrain the simulated LWP and AI. Current satellite observations of LWP and AI (e.g. MODIS, AATSR) show considerable differences. Without more observations to better

constrain LWP (or other cloud properties) and AI it is not clear which present day simulation (E5_Prog, E5_Ref, E6_Ref) is most realistic and which susceptibilities computed from these simulations (E5_Prog, E5_Ref, E6_Ref) are more realistic.

### 4.1.4 Impact of analysed region

Because buffering effects of aerosol-cloud interactions can depend on cloud type (Stevens and Feingold, 2009; Christensen et al., 2016) and some areas are affected by wet scavenging also in the non-raining scenes (see Fig. 2c), we compute next to

global mean values (between 60°N and 60°S) also mean values for two stratocumulus regions. The Californian stratocumulus region in the Northeast Pacific (15-55°N, 100-155°W) and the Peruvian stratocumulus region in the Southeast Pacific (10-45°S, 70-130°W), ocean only, are investigated. These are two regions where low liquid clouds and stable environmental regimes are frequent (see Fig. 1) and they are in general less affected by wet scavenging than regions in the tropics (see Fig. 1c). In Fig. 8 the $\alpha$ susceptibility is shown for both stratocumulus regions and all eight environmental

regimes. The $\alpha$ susceptibilities of the Californian stratocumulus region are similar to the global $\alpha$ susceptibilities in Fig. 5, whereas in the Peruvian stratocumulus region they are somewhat stronger for ECHAM6-HAM2. For AATSR-CAPA the $\alpha$ susceptibilities are stronger in both stratocumulus regions than globally, whereas for MODIS-CERES the $\alpha$ susceptibilities are similar in both stratocumulus regions and globally. Overall the $\alpha$ susceptibilities in the different analysed regions are qualitatively similar. The previous findings that the susceptibilities are weaker in the raining scenes than in the non-raining

scenes and that ECHAM6-HAM2 shows otherwise no strong dependence on environmental regime are qualitatively the same in the two stratocumulus regions. Similar results were found for the susceptibilities of other cloud properties (not shown). Restricting the analysis to low liquid clouds and the differentiation by environmental regimes seems therefore to be sufficient to separate different cloud types and the differentiation between raining and non-raining scenes seems to minimize the effect of wet scavenging for the non-raining scenes.

## 4.2 Effective radiative forcing

From the susceptibility of $\alpha$ to changes in AI the ERF$_{aci}$ can be estimated. Fig. 9 shows estimates of ERF$_{aci}$ for the low liquid clouds over global oceans analysed in this study. Not including aerosol water in the computation of AI leads to a much



weaker intrinsic ERF$_{aci}$ in ECHAM6-HAM2 in better agreement with estimates of AATSR-CAPA and MODIS-CAPA. Intrinsic ERF$_{aci}$ is stronger for non-raining scenes compared to the estimate for all scenes because wet scavenging of aerosol particles by precipitation is affecting the $\alpha$ susceptibility by removing more aerosols from clouds with a higher $\alpha$ (which are more likely to produce more precipitation) and thereby wet scavenging can lead to a weaker intrinsic ERF$_{aci}$ estimate. This indicates that the (strengthening) effect of aerosol swelling on $\alpha$ susceptibility to changes in aerosol is larger than the (weakening) effect of wet scavenging. This makes our best estimate for model intrinsic ERF$_{aci}$ of -0.4 W/m$^2$ for low liquid clouds over global oceans larger than the satellite data estimates. For most of the satellite data we have only estimates for all scenes but they are also likely affected by precipitation (which could even increase the difference in model vs. satellite based estimates). Chen et al. (2014) found slightly less negative values of intrinsic ERF$_{aci}$ of MODIS-CERES data for non-raining scenes than for all scenes. This mismatch in model and satellite ERF$_{aci}$ estimates could be an indication of missing or not well represented processes in ECHAM6-HAM2 like cloud top entrainment.

The estimates for extrinsic ERF$_{aci}$ on the contrary are smaller in ECHAM6-HAM2 than in AATSR-CAPA and MODIS-CAPA and are close to zero for the non-raining scene dry aerosol extrinsic ERF$_{aci}$ estimate in ECHAM6-HAM2. The changes in cloud cover are affected by aerosol swelling and other artefacts though (Quaas et al, 2010). Indeed the extrinsic ERF$_{aci}$ estimates are smaller and even positive for the dry aerosol in ECHAM6-HAM2 and also smaller when excluding near cloud aerosol in AATSR-CAPA and MODIS-CAPA. Chen et al. (2014) report also that using a smaller horizontal resolution for the analysis than was used in our study for MODIS-CERES leads to a smaller extrinsic ERF$_{aci}$ estimate which may be due to a scale problem (McComiskey and Feingold, 2012).

## 5 Summary and conclusions

It has been recognized in the scientific community that the statistical analysis of aerosol-cloud interactions can be affected by artefacts like cloud contamination or 3D-effects, by co-variations with relative humidity, by effects of clouds on aerosols like wet scavenging or aerosol processing, by absorbing aerosols or by differences in the analysis/observational scale and the process scale. Aerosol swelling has further been identified as the most likely reason for the large cloud cover susceptibility to changes in aerosol in satellite observations. Whereas the effect of aerosol swelling on the cloud cover and precipitation rate susceptibilities and how to minimize it has received attention in the literature, the effect on susceptibilities of other cloud variables is less explored. Our results with the global aerosol-climate model ECHAM6-HAM2 show that also the LWP and $\alpha$ and to a smaller extent also the CDNC are affected by aerosol swelling. By removing aerosol water (and therefore aerosol water uptake) from the computation of susceptibilities, the susceptibilities are considerably reduced and the 'dry' susceptibilities agree better with those of AATSR-CAPA and MODIS-CERES. For AATSR satellite data the effect of aerosol swelling is minimized by CAPA with a minimum distance between aerosol and cloud pixel. The MODIS AOD algorithm uses also a minimum distance between aerosol and cloud pixels and removes 25% of the brightest pixels. Although the hygroscopic growth of aerosol cannot be completely suppressed in the satellite data, due to the non-linearity of




hygroscopic growth we argue that when comparing to satellite data that minimize aerosol swelling it is better to use the dry aerosol of model simulations than the aerosol including aerosol water.

Our results show further that next to aerosol swelling, also wet scavenging and aerosol processing have an impact on LWP, $\alpha$ and CDNC susceptibilities. A separation in raining and non-raining scenes minimized the effect of wet scavenging for the non-raining scenes. For ECHAM6-HAM2 this separation was based on model precipitation as $R_e$ is not a good proxy for precipitation state when analysing aerosol-cloud interactions in ECHAM6-HAM2. Aerosol processing leads to negative LWP susceptibilities due to changes in AI in ECHAM6-HAM2, likely due to aerosol size changes by aerosol processing. The AOD is less dependent on aerosol size. Thus the LWP susceptibility to changes in AOD has fewer regions with negative LWP susceptibility even when aerosol processing is switched on in ECHAM6-HAM2. This is an indication that AOD, even though it does depend on aerosol mass rather than aerosol number, could be a better proxy for CCN than AI. This calls for further research on the effect of aerosol processing on the suitability of AOD and AI as proxies for CCN.

A simulation with a prognostic precipitation (rain, drizzle and snow) scheme in ECHAM5-HAM showed that the large LWP susceptibility cannot be explained by an overestimation of the CDNC dependent autoconversion. While using a prognostic precipitation scheme considerably reduces the ratio of autoconverison to autoconversion + accretion compared to a diagnostic precipitation scheme, it still leads to a large LWP susceptibility because the prognostic drizzle causes higher LWP and AI (variability) in stratocumulus regions compared to the diagnostic precipitation scheme.

A differentiation of susceptibilities by different environmental regimes (precipitation, stability in the lower troposphere, RH in the lower free troposphere) revealed that ECHAM6-HAM2 is less sensitive to different environmental regimes than AATSR-CAPA or MODIS-CERES satellite data (although also the two satellite datasets do not agree in their dependence on environmental regimes). The lack of sensitivity in ECHAM6-HAM2 is an indication that feedback processes like cloud top entrainment are missing or not well represented in the model.

**Data availability**

The Centre for Environmental Data Analysis (CEDA; http://www.ceda.ac.uk) provided the AATSR satellite data and NASA Goddard (https://ladsweb.nascom.nasa.gov) provided the MODIS satellite data used in this study. Model data is available from David Neubauer (david.neubauer@env.ethz.ch).

*Author contributions.* David Neubauer designed the analysis, conducted the simulations and computed susceptibilities for ECHAM-HAM and computed the effective radiative forcing estimates. Matthew Christensen computed susceptibilities for MODIS-CERES and MODIS-CAPA. Caroline Poulsen provided support needed to run ORAC. Ulrike Lohmann contributed to the analysis and interpretation of findings. David Neubauer prepared the manuscript with contributions from co-authors.

*Competing interests.* The authors declare that they have no conflict of interest.



*Acknowledgements.* This work was supported by the European Space Agency as part of the Aerosol_cci project (ESA Contract No. 4000109874/14/I-NB). We thank the ESA Cloud_cci project for providing the cloud satellite data. The ECHAM-HAMMOZ model is developed by a consortium composed of ETH Zurich, Max Planck Institut für Meteorologie, Forschungszentrum Jülich, University of Oxford, the Finnish Meteorological Institute and the Leibniz Institute for Tropospheric Research, and managed by the Center for Climate Systems Modeling (C2SM) at ETH Zurich. The Center for Climate Systems Modeling (C2SM) at ETH Zurich is acknowledged for providing technical and scientific support.

## Appendix

### Sampling effects

AATSR observations are done at a mean local solar time of 10:30am while for ECHAM6-HAM2 3-hourly instantaneous data is used. For ECHAM6-HAM2 data therefore the diurnal cycle of clouds and aerosol are resolved while AATSR data is always available at the same time. Resolving the diurnal cycle or not can potentially lead to a difference in the computed susceptibilities. To estimate the effect of the different sampling frequency and lacking temporal collocation (Schutgens et al., 2016) we compute the LWP susceptibility to changes in AI of a 17 year ECHAM6-HAM2 simulation one time from 3-hourly output and one time from data at 10:30 am, temporally collocated with AATSR. The results are shown in Fig. A. The maxima and minima of the LWP susceptibility are more pronounced with the 10:30am local time sampling than with the 3-hourly sampling. The general geographical pattern and magnitude of the LWP susceptibility is quite similar between the two sampling methods. As the global ECHAM6-HAM2 simulations have to use a relatively coarse resolution (T63, $1.9° \times 1.9°$), high temporal sampling is necessary to obtain enough aerosol-cloud data pairs to compute significant linear regressions, in particular as we differentiate environmental regimes compute susceptibilities at the native model grid to reduce effects of aggregation (Grandey and Stier, 2010; McComiskey and Feingold, 2012). As the benefits of the analysis of different environmental regimes with reduced aggregation effects outweighs the difference due to resolving the diurnal cycle or not and lack of temporal collocation, we have chosen the 3-hourly instantaneous data for our analysis.

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





Table 1. Minimum and maximum values for aerosol and cloud properties used in this study. AOD is aerosol optical depth, AI is aerosol index, CDNC is cloud droplet number concentration, LWP is liquid water path, COD is cloud optical depth, $R_e$ is cloud droplet effective radius. CDNC and $R_e$ are cloud top values.

| Variable | Minimum value | Maximum value |
|---|---|---|
| AOD | 0.01 | 1 |
| AI | 0.02 | 0.6 |
| CDNC ($\#/cm^3$) | 1 | 1000 |
| LWP ($g/m^2$) | 1 | 1000 |
| COD | 0.1 | 300 |
| $R_e$ (µm) | 1 | 50 |



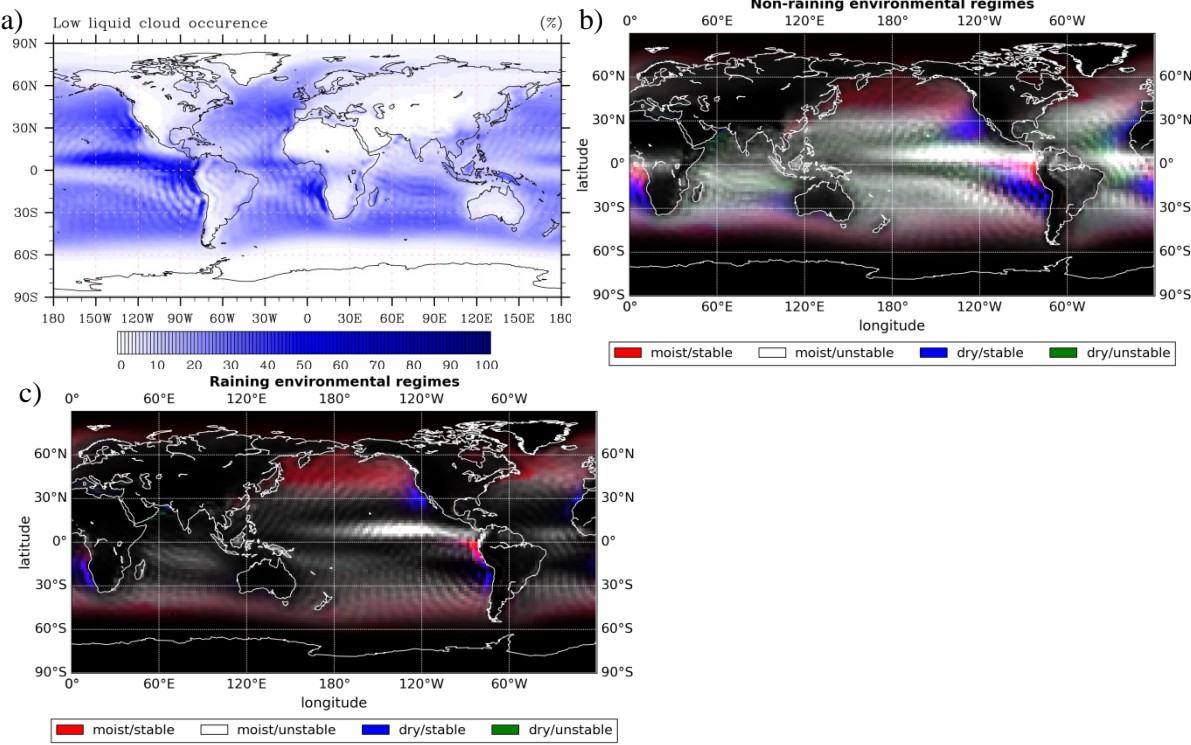

**Figure 1**: Average frequency of the occurrence of low liquid clouds (cloud top pressure > 500 hPa, cloud top temperature > 273.15 K) in E6_Ref between 1995 and 2012 in a) for all environmental regimes together, b) for non-raining regimes, c) for raining regimes.









**Figure 2:** Susceptibility of LWP to changes in AI or AOD for ECHAM6-HAM2 (E6_Ref and E6_AProc) when low liquid clouds and aerosol are present during the simulation period 1995-2012 between 60°N and 60°S. a) response in E6_Ref to AI for all scenes, b) the same as in a) but for dry AI, c) same as in b) but only for non-raining scenes (precipitation < 0.5 mm/day), d) same as in c) but with a different definition for non-raining scenes ($R_e$ < 14 μm), e) same as in c) but for E6_AProc, f) same as in e) but for dry AOD instead of dry AI, g) same as in f) but for E6_Ref.

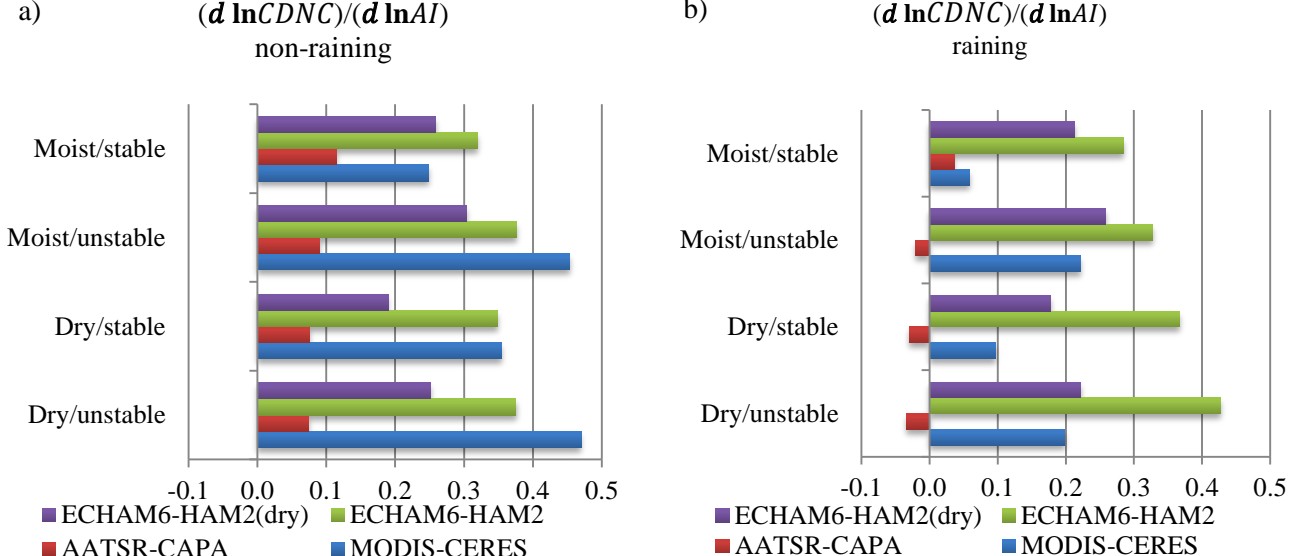

**Figure 3:** Susceptibility of CDNC to changes in AI for ECHAM6-HAM2 (E6_Ref), E6_Ref without aerosol water uptake (dry) during 1995-2012, for AATSR-CAPA using the full satellite record span 2002-2012 and for MODIS-CERES during 2006-2010. The definition of the different environmental regimes is given in the text. a) For all non-raining scenes, b) for all raining scenes. Only values over oceans are shown. The MODIS-CERES data is from Christensen et al. (2016).



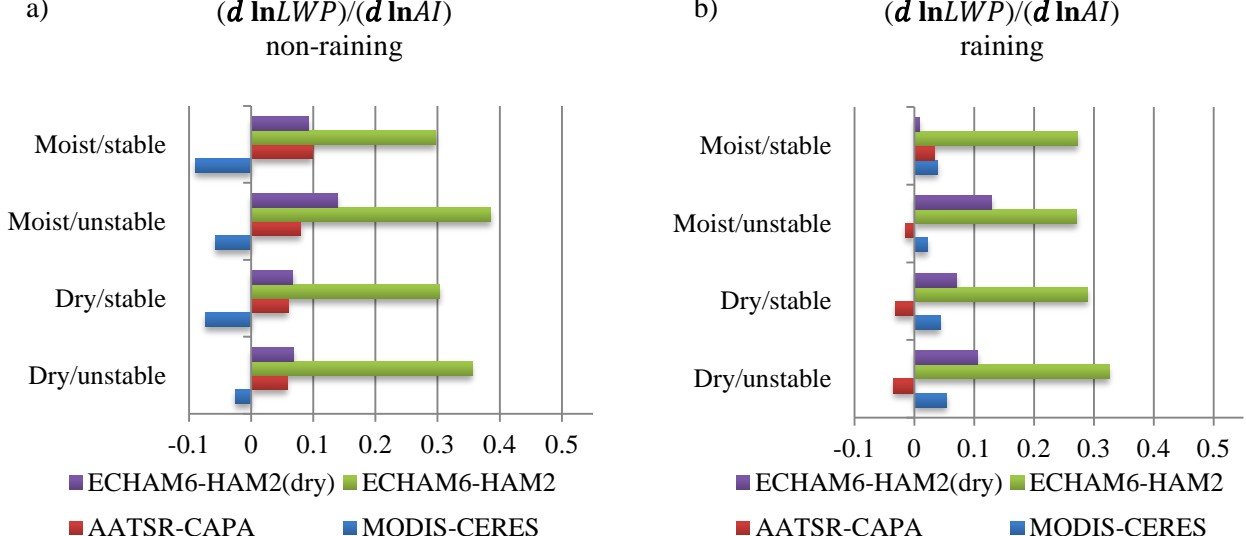

**Figure 4:** Same as Fig. 3 but for the LWP susceptibility to changes in AI for ECHAM6-HAM2 (E6_Ref), E6_Ref without aerosol water uptake (dry), AATSR-CAPA and MODIS-CERES. The MODIS-CERES data is from Christensen et al. (2016).

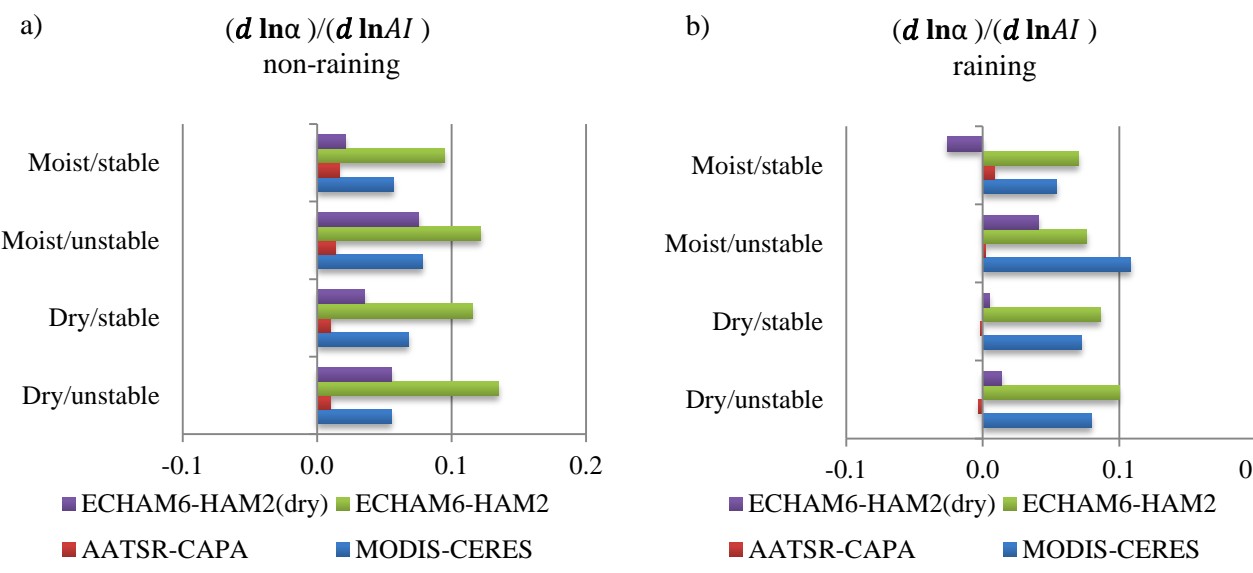



**Figure 5:** Same as Fig. 3 but for the shortwave cloud albedo susceptibility to changes in AI for ECHAM6-HAM2 (E6_Ref), E6_Ref without aerosol water uptake (dry), AATSR-CAPA and MODIS-CERES. The MODIS-CERES data is from Christensen et al. (2016).

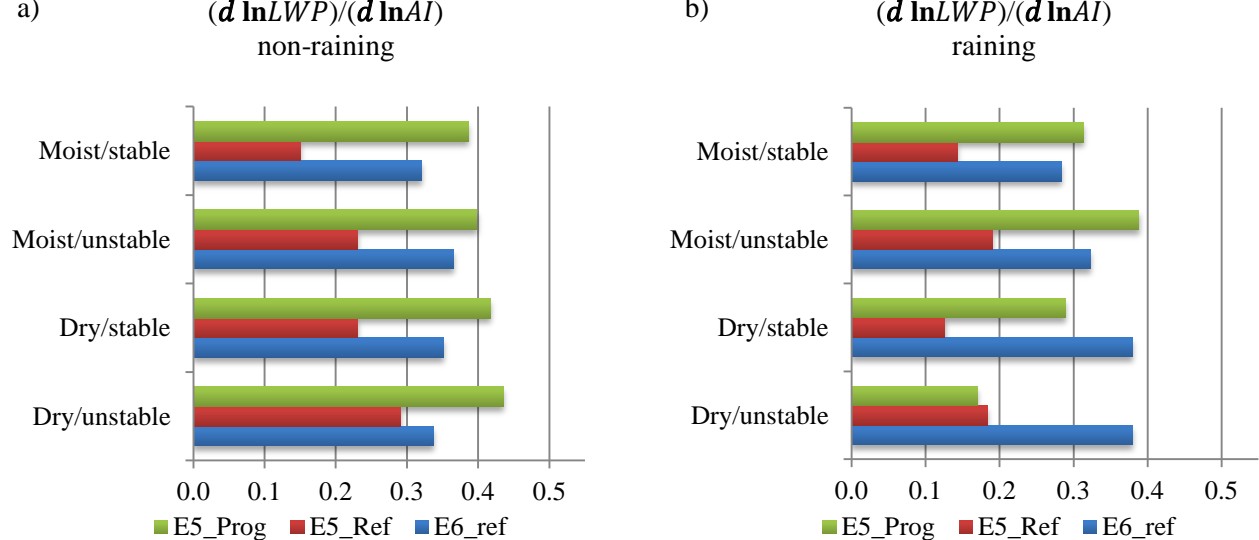

**Figure 6:** Same as Fig. 3 but for the LWP susceptibility to changes in AI for E5_Prog, E5_Ref and E6_Ref.





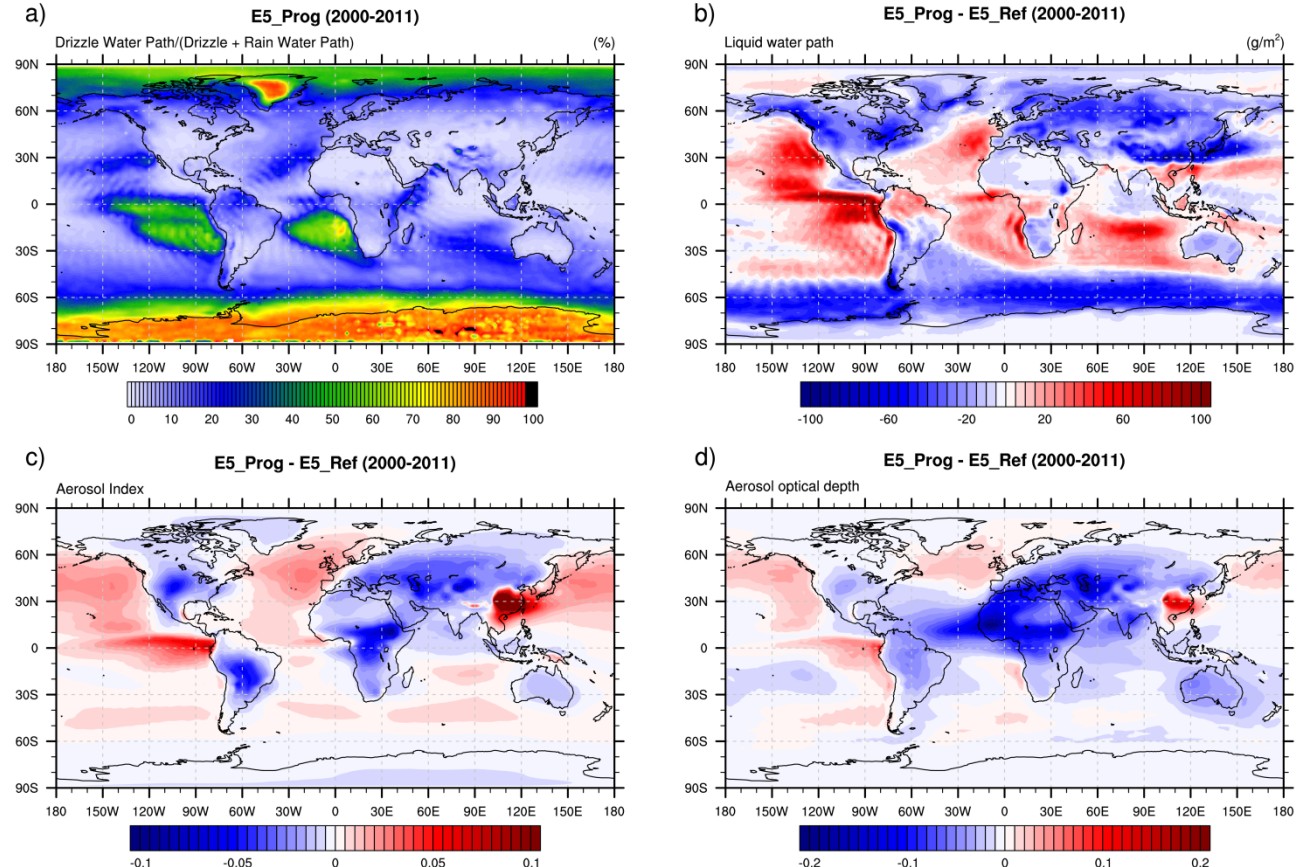

**Figure 7:** a) Annual mean ratio of drizzle water path to the sum of drizzle and rain water path for the E5_Prog simulation. The difference between E5_Prog and E5_Ref for 12 years of simulations (2000-2011) b) LWP, c) AI, d) AOD. a) and b) include precipitation and LWP from all clouds not only low liquid clouds, c) and d) include cloudy and cloud-free scenes.





**Figure 8:** Same as Fig. 5 but for the shortwave cloud albedo susceptibility to changes in AI for ECHAM6-HAM2 (E6_Ref), E6_Ref without aerosol water uptake (dry), AATSR-CAPA and MODIS-CERES in the (a,b) Californian and (c,d) Peruvian stratocumulus regions. Not enough aerosol-cloud data pairs are available in the stratocumulus regions that the linear regression is significant for the dry/unstable regimes in ECHAM6-HAM2 except for the non-raining scenes in the Peruvian region. The MODIS-CERES data is from Christensen et al. (2016).





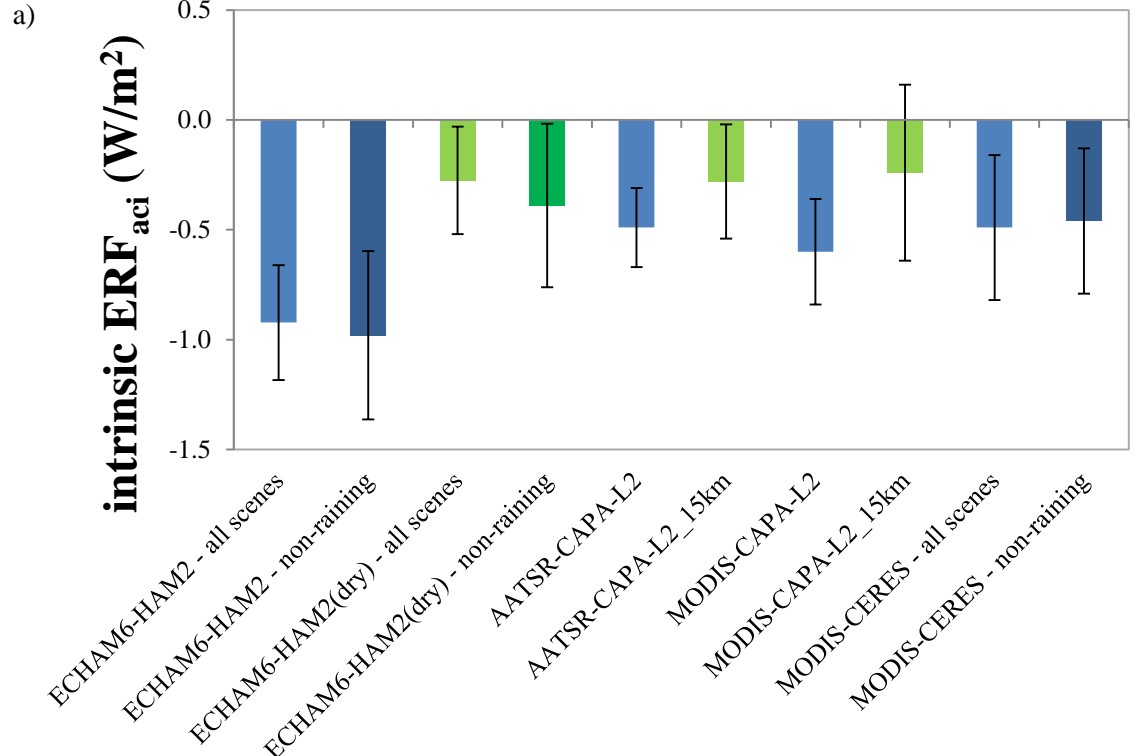





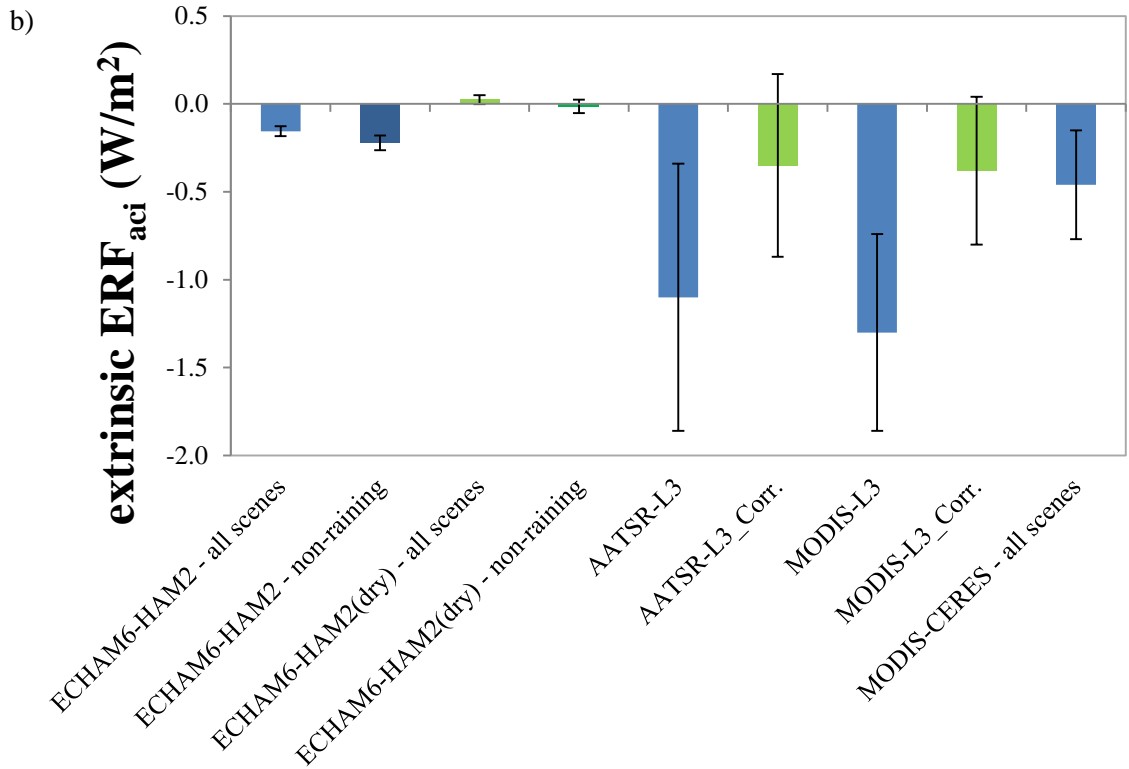

**Figure 9:** Estimate of a) intrinsic and b) extrinsic effective radiative forcing due to aerosol-cloud interactions (ERF$_{aci}$) of low liquid clouds between 60°N and 60°S. For ECHAM6-HAM2 separate estimates for all scenes and non-raining scenes (< 0.5 mm / day) as well as with/without aerosol water uptake (dry aerosol) are shown. For AATSR-CAPA and MODIS-CAPA estimates with all aerosol particles (L2/L3) and excluding near cloud aerosol particles (L2_15km/L3_Corr.) are shown. Only values over oceans are shown. The AATSR-CAPA and MODIS-CAPA forcing data are from Christensen et al. (2017). Note that the values for MODIS-CAPA/MODIS-L3/MODIS-L3_Corr. are computed from three months of data for June, July and August 2008 only. The MODIS-CERES forcing data are from Chen et al. (2014). The uncertainty is based on the standard error of the linear regression. Light blue bars are used for all scenes with aerosol water uptake or including near cloud aerosol particles. Green bars indicate removal of aerosol water or near cloud aerosol particles. Dark bar colours are used for non-raining scenes.



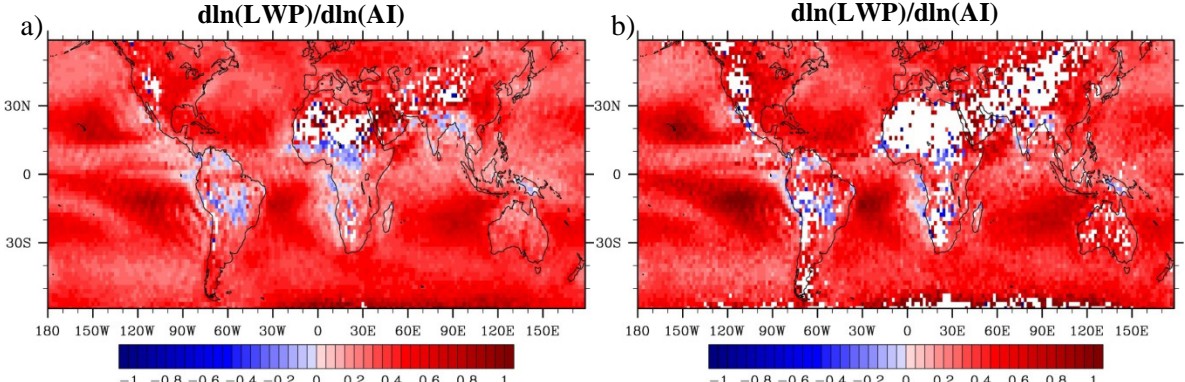

**Figure A:** LWP response to changes in AI for ECHAM6-HAM2 (1995-2011). a) For 3-hourly sampling, b) for daily
(10:30am local time) sampling.