# Peer review of "Unveiling aerosol-cloud interactions Part 2: Minimizing the effects of aerosol swelling and wet scavenging in ECHAM6-HAM2 for comparison to satellite data"

_Atmospheric Chemistry and Physics, 2017_

## Referee Comment (RC1) · Anonymous Referee #1 · 1 Jul 2017

This illuminating study helps to resolve previous disparities between simulated and observed relationships between clouds and aerosols. I particularly appreciate the physical mechanisms put forth to explain the different relationships under different assumptions. The combination of results for different model configurations is very helpful, and tells a compelling story.

Page 4, line 12. Eqs. (7) and (10) should be Eqs. (6) and (9).

Page 6 line 12. Replace "divided by to" with "divided by".

[Figure]

Page 6 line 20. Move "multiple linear regression could be used in principle" to the front of the sentence.

Page 6, line 31. How is AODaerosol water calculated? A better way would be to calculate AOD of the dry aerosol given its size and dry composition. It would help the reader to know how AOD is determined from the aerosol components.

Page 8, lines 22-25. Should note here the lower bound on droplet number.

Page 9, lines 1-7. Please explain how the aerosol processing scheme differs from configurations without it. Surely all configurations treat aqueous chemistry and nucleation scavenging in some manner, right?

Page 10, line 8. Relative to what? Why not be quantitative? Say, "exceeds 0.8 in many areas".

Page 11, line 15. Make it clear that figure 2g is without aerosol processing.

Page 11, lines 15-16. How is this statement support by the results? CCN depends on particles that do not contribute much to AOD, so why should AOD be better than AI? I think what you mean to say is AI includes the effects of aerosol processing, while AOD isolates CCN effects on cloud before cloud processing (line 14). I don't agree with that statement; you can't isolate processes when interactions are strong; you have to look at relationships between the variables that control the processes, which is why CCN is best.

Page 11, line 27. Insert "averaged" before "over". Figure 3 caption should make this clear.

Page 11, lines 27-32. Why not discuss AATSR-CAPA and MODIS-CERES results here?

Page 12, line 1. Make it clear this is averaged over the oceans.

Page 12 lines 34-35. "Also" used twice.

Page 13, line 24. New paragraph.

Page 16, lines 8-11. Again, I question this conclusion. Aerosol processing is an important part of cloud-aerosol interactions.

---

## Referee Comment (RC2) · Anonymous Referee #2 · 31 Jul 2017

This work looks at different factors that can affect the AI-LWP relationship, from measurement issues such as aerosol humidification to differences in how models represent aerosol and cloud processes. The authors find that model processes, such as wet scavenging, the use of prognostic drizzle and the representation of cloud processing of aerosol can have a significant effect on the AI-LWP susceptibility. They suggest that the susceptibility of LWP to dry aerosol properties is a better way to compare models to observations, as long as the satellite observations are sampled in a way that can reduce the impact of aerosol humidification. They go on to note that the differences

between the MODIS and AATSR relationships mean that current satellite relationships are problematic for use constraining the strength of aerosol-cloud interactions in global models.

The subject of this paper would be of interest to the readers of Atmospheric Chemistry and Physics, looking at observational constraints on aerosol indirect effects in global climate models. It provides an useful comparison between model and satellite relationships and I think that with a few minor changes/clarifications it would be suitable for publication.

Minor points

P1L23: This is a very long sentence and the meaning is not quite clear

P3L22: While vertical information is nice to have, other studies suggest that it may not be required to achieve s good proxy for CCN, both Stier (2016) and Gryspeerdt et al., (2017) find that AI is a good proxy for CCN (or is able to diagnose PD-PI CDNC changes), despite being vertically integrated.

P3L29: linearly

P4L29: Presumably this is for the model, as the MODIS LWP/CDNC can only be calculated in daylight for observations

P5L26: The MODIS aerosol retrieval is not performed poleward of 60 degrees anyway

P7L9: While it may be true that the sensitivities are of a similar magnitude, if the AI perturbation has a different magnitude to the AOD perturbation, these two relationships will diagnose different changes in albedo. Just because the relationships are a similar magnitude does not mean they are interchangeable.

P8L15: 'is an aerosol-climate model ... only the aerosol-climate model part is used.' - At the moment this sentences does not say much, is it missing something?

P9L25: Cloud top pressures less than 500hPa - how are these selected from the model,

is a satellite simulator used?

P10L30: Is this use of Re as a proxy for precipitation dependent on the cloud parametrisation? Is it known if the ECHAM parametrisation is theoretically capable of this kind of behaviour?

P11L10: Presumably this influence of cloud processing could be checked within the model? Or if the effect is known, it could be stated more strongly.

P11L14: I am not sure I understand the reasoning here (and this is an important point) as to why AODdry is a better proxy than AIdry? AODdry is less sensitive to aerosol size than AIdry, but aerosol activation is quite sensitive to aerosol size.

P11L27: Although the meteorological regimes are a good way to look at this, the split by humidity regimes may also confound different cloud or aerosol types. Maps of these sensitivities might be useful (at the authors' discretion)

P11L31: The AI-CDNC relationship is mainly looking at aerosol activation - does wet scavenging really affect this, or is the change in the relationship in precipitating scenes indicative of differing aerosol types/cloud updraughts?

P12L2: based on Fig. 4a, I would have said that the regime variability in ECHAM using AIdry is similar, or even larger than the satellite products.

P12L19: Is there a way of checking if sampling is the issue here? Are there some situations where MODIS/AATSR refuse to retrieve cloud/aerosol properties?

P12L30: Does alpha not depend on the cloud properties to some extent (if not these retrieved ones), when computing the fluxes from CERES broad-band radiances? Perhaps this is not a significant issue?

P13L25: Fig. 7a shows drizzle water path, rather than LWP

P13L29: This is not true for all relationships (e.g. Gryspeerdt et al., 2017). This might just mean that the AI-LWP relationship is not a good proxy for the strength of the

aerosol influence on LWP.

P14L16: Could these regions be drawn on the maps (perhaps in fig 1)

P15L1: Could these ERFaci values be compared with values determined from the model (PD-PI simulations)?

P16L20: See earlier comment about model vs. satellite variability (P12L3)

References

Gryspeerdt, E. et al. (2017), Constraining the instantaneous aerosol influence on cloud albedo, Proc. Natl. Acad. Sci. USA, 114(19), 4899–4904, doi:10.1073/pnas.1617765114.

Stier, P. (2016), Limitations of passive remote sensing to constrain global cloud condensation nuclei, Atmos. Chem. Phys., 16(10), 6595–6607, doi:10.5194/acp-16-6595-2016.

---

## Author Comment (AC1) · 19 Sep 2017

**Response to comment on *Unveiling aerosol-cloud interactions Part 2: Minimizing the effects of aerosol swelling and wet scavenging in ECHAM6-HAM2 for comparison to satellite data*, Atmos. Chem. Phys. Discuss., https://doi.org/10.5194/acp-2017-449**

David Neubauer, Matthew W. Christensen, Caroline Poulsen, Ulrike Lohmann

We would like to thank the reviewer for the helpful comments and suggestions. They have helped to improve the content of the paper.
The original comments are in black. Responses are in blue. *Modifications to the text are in green and italics*.

**Anonymous Referee #1**

This illuminating study helps to resolve previous disparities between simulated and observed relationships between clouds and aerosols. I particularly appreciate the physical mechanisms put forth to explain the different relationships under different assumptions. The combination of results for different model configurations is very helpful, and tells a compelling story.

Thank you for this encouraging assessment and your valuable comments and suggestions to improve the manuscript. The anthropogenic CCN increase used in the computation of the forcing estimates was changed in the revised manuscript, which has a large impact on the forcing values. The anthropogenic CCN increase is now estimated from AI instead of AOD changes (from simulations with present day and pre-industrial aerosol emissions). Although a disparity between the simulated and observed $ERF_{aci}$ is present in the revised manuscript the overall conclusions remain valid.

Page 4, line 12. Eqs. (7) and (10) should be Eqs. (6) and (9).
Done.

Page 6 line 12. Replace "divided by to" with "divided by".
Done.

Page 6 line 20. Move "multiple linear regression could be used in principle" to the front of the sentence.
Done.

Page 6, line 31. How is AODaerosol water calculated? A better way would be to calculate AOD of the dry aerosol given its size and dry composition. It would help the reader to know how AOD is determined from the aerosol components.
AODaerosol water is calculated by weighting AOD with the volume fraction of aerosol water. We agree that it would be better to calculate AOD of the dry aerosol from its size distribution and dry composition. Unfortunately, the necessary diagnostic is not available. We do not expect a change in the qualitative results i.e. that cloud variables are less susceptible to changes in AIdry than AI by using this approximation (or less to AODdry than to AOD).

*$AODdry = AOD - AOD_{aerosol\,water} =$*
*$AOD \times (1 - volume_{aerosol\,water}/volume_{total\,aerosol})$ (1)*
*$AOD_{aerosol\,water}$ is calculated by multiplying AOD by the volume fraction of aerosol water ($volume_{aerosol\,water}/volume_{total\,aerosol}$). All aerosol particles are assumed to be spherical in this calculation.*

Page 8, lines 22-25. Should note here the lower bound on droplet number.
Done.

*A minimum cloud droplet number concentration of 40/cm$^3$ is used in ECHAM6-HAM2 and 20/cm$^3$ in ECHAM5-HAM.*

Page 9, lines 1-7. Please explain how the aerosol processing scheme differs from configurations without it. Surely all configurations treat aqueous chemistry and nucleation scavenging in some manner, right?
The description of the aerosol processing scheme has been expanded. The main difference to the standard configuration is that the aerosol masses of the different aerosol species in cloud droplets and ice crystals are prognostic variables and that these masses are traced throughout all processes (nucleation, collisions, evaporation, aqueous chemistry, …). These processes are also computed in the standard configuration but there the aerosol is simply removed or added to the interstitial aerosol at the end of each timestep.

*ECHAM-HAM in its standard configuration does not track aerosol particles in hydrometeors. In the standard configuration scavenged aerosol particles (by nucleation and/or impaction scavenging) are removed from the interstitial aerosol (evaporation of rain or sublimation of snow below cloud base release part of the scavenged aerosol particles back to the atmosphere though) and sulphate produced by heterogeneous chemistry is added to the interstitial aerosol. With the aerosol processing scheme on the other hand, aerosol mass transfers to and from in-cloud aerosol tracers by nucleation and impact scavenging, freezing and evaporation of cloud droplets, and melting and sublimation of ice crystals are tracked. These processes are computed explicitly. Sulphate produced by heterogeneous chemistry is added to the in-cloud sulphate aerosol tracer. Aerosol particles from evaporating/sublimating clouds and precipitation are released to the modes that correspond to their size with the aerosol processing scheme.*

Page 10, line 8. Relative to what? Why not be quantitative? Say, "exceeds 0.8 in many areas".
Agreed. We changed this sentence to:

*The LWP susceptibility is positive almost everywhere (i.e. an increase in AI leads to an increase in LWP and a decrease in AI leads to a decrease in LWP) and the LWP susceptibility exceeds 0.5 in many areas.*

Page 11, line 15. Make it clear that figure 2g is without aerosol processing.
Done.

*AODdry is less sensitive to aerosol size than AIdry so the negative LWP susceptibility shown in Fig. 2e should rather be due to changes in aerosol size than in aerosol number or mass (for comparison the LWP susceptibility to changes in AODdry of E6_Ref (i.e. without aerosol processing) is shown in Fig. 2g).*

Page 11, lines 15-16. How is this statement support by the results? CCN depends on particles that do not contribute much to AOD, so why should AOD be better than AI? I think what you mean to say is AI includes the effects of aerosol processing, while AOD isolates CCN effects on cloud before cloud processing (line 14). I don't agree with that statement; you can't isolate processes when interactions are strong; you have to look at relationships between the variables that control the processes, which is why CCN is best.

This statement was not well formulated and AOD should indeed not be a better proxy for CCN than AI (Nakajima et al., 2001) because, as you point out, AOD does not correlate well with aerosol number. We therefore removed this statement from the text and abstract and only point out the need to investigate the effect of aerosol processing on this kind of statistical relationships.

*Further research for example using a bin representation of aerosol size could give further insight of the effect of aerosol processing on aerosol-cloud interactions.*

Page 11, line 27. Insert "averaged" before "over". Figure 3 caption should make this clear.

A sentence at the beginning of subsection 1.4.2 was added to make clear that only grid boxes over the global oceans are analysed. "over oceans" was replaced by "averaged over global oceans" in subsection 1.4.2 and captions of Figure 3 and 9.

*In the weighted averaging only grid boxes over the global oceans are taken into account.*

Page 11, lines 27-32. Why not discuss AATSR-CAPA and MODIS-CERES results here?

We wanted to focus on the difference between AI and AIdry for the CDNC susceptibility and therefore only discussed ECHAM6-HAM2 results. But as we discuss AATSR-CAPA and MODIS-CERES results for the other susceptibilities it is more consistent to add them for the CDNC susceptibility as well. Therefore, the discussion of AATSR-CAPA and MODIS-CERES results was added to the discussion of ECHAM6-HAM2 results.

*For ECHAM6-HAM2, AATSR-CAPA and MODIS-CERES the CDNC susceptibility to AI varies only little between moist or dry free tropospheric conditions and a stable or unstable lower troposphere. The CDNC susceptibility of ECHAM6-HAM2 to AIdry is generally smaller, up to 50% less depending on the regime. The CDNC susceptibility of AATSR-CAPA is smaller than for MODIS-CERES or ECHAM6-HAM2 (AI or AIdry). The minimum distance of the CAPA-algorithm should reduce the effects of aerosol swelling, cloud contamination and 3D radiative effects by selecting aerosols farther away from clouds where these satellite artefacts should be minimal. For AATSR-CAPA this seems to lead to a small CDNC susceptibility. For ECHAM6-HAM2 and MODIS-CERES the differences between non-raining and raining scenes are small and in general the CDNC susceptibility is smaller in the raining scenes than in the non-raining scenes which is an indication of wet scavenging affecting aerosol concentrations in the raining scenes. For AATSR-CAPA the CDNC susceptibility to AI is smaller in the moist stable regime in the raining than in the non-raining scenes and even negative in the other regimes in the raining scenes, also indicative of wet scavenging in the raining scenes.*

Page 12, line 1. Make it clear this is averaged over the oceans.

Done.

*The response of LWP to changes in AI (dlnLWP/dlnAI), averaged over the global oceans, shown in Fig. 4, reveals larger susceptibilities and lower variability in susceptibilities between environmental regimes in ECHAM6-HAM2 than in satellite observations.*

Page 12 lines 34-35. "Also" used twice. Page 13, line 24. New paragraph.
Both done.

Page 16, lines 8-11. Again, I question this conclusion. Aerosol processing is an important part of cloud-aerosol interactions.
See our response to your comment above. We removed the statement that AOD could be a better CCN proxy than AI and only point out the need to investigate the effect of aerosol processing on this kind of statistical relationships.

*This calls for further research on the effect of aerosol processing when analysing the effects of changes in CCN on cloud properties.*

---

## Author Comment (AC2) · 19 Sep 2017

**Response to comment on *Unveiling aerosol-cloud interactions Part 2: Minimizing the effects of aerosol swelling and wet scavenging in ECHAM6-HAM2 for comparison to satellite data*, Atmos. Chem. Phys. Discuss., https://doi.org/10.5194/acp-2017-449**

David Neubauer, Matthew W. Christensen, Caroline Poulsen, Ulrike Lohmann

We would like to thank the reviewer for the helpful comments and suggestions. They have helped to improve the content of the paper.
The original comments are in black. Responses are in blue. *Modifications to the text are in green and italics.*

**Anonymous Referee #2**

This work looks at different factors that can affect the AI-LWP relationship, from measurement issues such as aerosol humidification to differences in how models represent aerosol and cloud processes. The authors find that model processes, such as wet scavenging, the use of prognostic drizzle and the representation of cloud processing of aerosol can have a significant effect on the AI-LWP susceptibility. They suggest that the susceptibility of LWP to dry aerosol properties is a better way to compare models to observations, as long as the satellite observations are sampled in a way that can reduce the impact of aerosol humidification. They go on to note that the differences between the MODIS and AATSR relationships mean that current satellite relationships are problematic for use constraining the strength of aerosol-cloud interactions in global models.
The subject of this paper would be of interest to the readers of Atmospheric Chemistry and Physics, looking at observational constraints on aerosol indirect effects in global climate models. It provides an useful comparison between model and satellite relationships and I think that with a few minor changes/clarifications it would be suitable for publication.

Thank you for your insightful comments and suggestions to improve the manuscript. The suggested change in the calculation of the anthropogenic aerosol increase has significantly changed $ERF_{aci}$ from ECHAM-HAM2, with unrealistically large values for the humid aerosol and therefore further strengthens the arguments for the use of dry aerosol.

Minor points
P1L23: This is a very long sentence and the meaning is not quite clear
This sentence was split into three sentences and it was specified what the disagreements between the datasets are to make this statement clear.

*We further find that the statistical relationships inferred from different satellite sensors (AATSR-CAPA vs. MODIS-CERES) as well as from ECHAM6-HAM2 are not always of the same sign for the tested environmental conditions. In particular the susceptibility of the liquid water path is negative in non-raining scenes for MODIS-CERES but positive for AATSR-CAPA and ECHAM6-HAM2. Feedback processes like cloud top entrainment that are missing*

*or not well represented in the model are therefore not well constraint by the satellite observations.*

P3L22: While vertical information is nice to have, other studies suggest that it may not be required to achieve s good proxy for CCN, both Stier (2016) and Gryspeerdt et al., (2017) find that AI is a good proxy for CCN (or is able to diagnose PD-PI CDNC changes), despite being vertically integrated.

We agree that the results of Gryspeerdt et al. (2017) show that AI is a better CCN proxy as AOD and that including vertical information is not that beneficial for most analysed models. However, Gryspeerdt et al. (2017) used CCN at 1 km altitude compared to column-integrated CCN to estimate the impact of vertical information whereas Stier (2016) analysed among others correlations between AI and CCN at cloud base. The correlations between AI and CCN at cloud base (Fig. 8 in Stier, 2016) are low e.g. in marine stratocumulus regions which are important regions for radiative effects of aerosol-cloud interactions. Therefore, we keep the original text as is but add results of Gryspeerdt et al. (2017).

*Gryspeerdt et al. (2017) showed that including vertical information is beneficial for several global aerosol-climate models but these benefits are smaller than using AI instead of AOD as a CCN proxy for most analysed models. The simulations by Stier (2016), Gryspeerdt et al. (2017) and surface measurements do not account for aerosol processing in clouds, which could affect the suitability of these aerosol quantities as CCN proxy.*

P3L29: linearly
Done.

P4L29: Presumably this is for the model, as the MODIS LWP/CDNC can only be calculated in daylight for observations
Indeed. This is now stated explicitly.

*… (this is only relevant for ECHAM6-HAM2 as the satellite retrievals are done for daylight scenes).*

P5L26: The MODIS aerosol retrieval is not performed poleward of 60 degrees anyway
For this reason, we excluded high latitudes from our analysis (high zenith angle, bright surfaces).

P7L9: While it may be true that the sensitivities are of a similar magnitude, if the AI perturbation has a different magnitude to the AOD perturbation, these two relationships will diagnose different changes in albedo. Just because the relationships are a similar magnitude does not mean they are interchangeable.
Thank you for this excellent point. We recomputed the $ERF_{aci}$ estimates using the anthropogenic aerosol increase calculated from AI and the $ERF_{aci}$ estimates increased significantly. We use therefore these new values in the manuscript and added a brief discussion of the impact of using AOD or AI for calculating the anthropogenic aerosol increase. We also added the comparison to $ERF_{aci}$ diagnosed from model simulations that you suggested. The overall conclusions remain valid and the unrealistically large $ERF_{aci}$ values for humid aerosol are a further argument for using dry aerosol for this kind of analysis.

Subsection 2.3:

..., $\Delta a_{AI} = ln \frac{AI}{AI - AI_{anth}}$ represents the anthropogenic aerosol increase ($AI_{anth}$ is anthropogenic AI), which is taken from reference model simulations (Neubauer et al, 2014) for ECHAM6-HAM2. Note that $\Delta a_{AOD}$ based upon AOD has been used in several studies (e.g. Quaas et al., 2008; Bellouin et al., 2013; Chen et al., 2014) therefore we compute Eq. (14) as a sensitivity test also with $\Delta a_{AOD}$ instead of $\Delta a_{AI}$.

and:

As a reference forcing for ECHAM6-HAM2, $ERF_{aci}$ was also diagnosed for low liquid clouds (cloud top pressures > 500 hPa and cloud top temperatures > 273.15 K) from simulations with present day and pre-industrial aerosol emissions.

Subsection 4.2:
For ECHAM6-HAM2, $ERF_{aci}$ was also diagnosed for low liquid clouds from simulations with present day and pre-industrial aerosol emissions. The thus diagnosed forcing of -0.7 W/m2 serves as a reference for ECHAM6-HAM2. Not including aerosol water in the computation of AI leads to a much weaker intrinsic+extrinsic $ERF_{aci}$ in ECHAM6-HAM2 (-0.8 W/m$^2$ for all scenes and -1.5 W/m$^2$ for non-raining scenes) in better agreement with the diagnosed reference forcing. The estimates of intrinsic+extrinsic $ERF_{aci}$ in ECHAM6-HAM2 when aerosol water is included are unrealistically large (-3.5 W/m$^2$ for all scenes and -4.5 W/m$^2$ for non-raining scenes) which shows the need to remove aerosol water when computing forcing estimates from present day variability. The results in Ghan et al. (2016) show an underestimation of cloud optical depth and cloud cover susceptibilities computed from present day variability compared to those computed from anthropogenic emissions. Our results for ECHAM6-HAM2 show in contrast to this a stronger intrinsic+extrinsic $ERF_{aci}$ (based on present day variability) compared to the diagnosed $ERF_{aci}$ (based on anthropogenic emissions). A reason for this may be that AI is a vertically integrated quantity that does not take the location of aerosol particles in the vertical nor their chemical composition into account (Gryspeerdt et al, 2017).
Not including aerosol water leads to a better agreement of intrinsic $ERF_{aci}$ of ECHAM6-HAM2 with estimates of AATSR-CAPA and MODIS-CAPA than when aerosol water is included but the model still shows considerably larger values of intrinsic $ERF_{aci}$ than the satellite estimates.

and:
The considerably larger estimates of intrinsic+extrinsic $ERF_{aci}$ in ECHAM6-HAM2 when aerosol water is included compared to previous studies (e.g. Quaas et al., 2008; Bellouin et al., 2013; Chen et al., 2014) are likely due to the use of different variables for the anthropogenic aerosol increase (i.e. AOD vs. AI). We recomputed $ERF_{aci}$ using $\Delta a_{AOD}$ (17% increase in global annual mean from pre-industrial) instead of $\Delta a_{AI}$ (44% increase in global annual mean from pre-industrial aerosol). The estimates of intrinsic+extrinsic $ERF_{aci}$ in ECHAM6-HAM2 are then much smaller (-1.1 W/m$^2$ for all scenes and -1.2 W/m$^2$ for non-raining scenes when aerosol water is included and -0.3 W/m$^2$ for all scenes and -0.4 W/m$^2$ for non-raining scenes when aerosol water is removed). This shows how important it is which variable is used to compute the anthropogenic aerosol increase (as anthropogenic aerosol particles are on average smaller than natural aerosol particles). This is in agreement with results of Gryspeerdt et al. (2017). A comparison of their Figs. 3a and 3b indicates also much weaker values for the anthropogenic aerosol increase computed from AOD than from AI or other proxies for the increase in CDNC.

P8L15: 'is an aerosol-climate model ... only the aerosol-climate model part is used.' - At the moment this sentences does not say much, is it missing something?

The full sentence in the online available discussion paper reads: "ECHAM-HAMMOZ is a global aerosol-chemistry climate model of which in this study only the global aerosol-climate model part is used." i.e. the sophisticated chemistry module MOZ is not used in this study.

P9L25: Cloud top pressures less than 500hPa - how are these selected from the model, is a satellite simulator used?

The selection of cloud top pressure > 500 hPa as well as cloud top temperature > 273.15 K is done offline from 3-hourly instantaneous output. It is now added.

*To focus only on warm, liquid clouds in the analysis, model cloud top pressure and temperature (from the 3-hourly instantaneous output) are used to identify low liquid clouds as those with cloud top pressures greater than 500 hPa and cloud top temperatures exceeding 273.15 K.*

P10L30: Is this use of Re as a proxy for precipitation dependent on the cloud parametrisation? Is it known if the ECHAM parametrisation is theoretically capable of this kind of behaviour?

These are interesting questions. The autoconversion and accretion parameterizations in ECHAM6-HAM2 follow Khairoutdinov and Kogan (2000). Khairoutdinov and Kogan (2000) developed their parameterizations for marine stratocumulus clouds using a drop spectrum resolving microphysical model. They mentioned though that the autoconversion rate varies more than two orders of magnitude when the mean volume radius changes from 7 to 19 μm. Although this indicates that the parameterizations in ECHAM6-HAM2 could make the model well capable for testing Re as a proxy for precipitation, we nevertheless mention that this result may depend on the used parameterizations.

*The differences shown in Fig. 2b and Fig. 2c may depend on the parameterizations used for precipitation formation (Suzuki et al., 2011) and also the tuning of these parameterizations (Suzuki et al., 2013). Further studies (e.g. with high resolution models) will be necessary to assess the usability of Re in a global model as a proxy for precipitation or the absence hereof.*

P11L10: Presumably this influence of cloud processing could be checked within the model? Or if the effect is known, it could be stated more strongly.

This is also an interesting question but such a check would be not trivial. It would involve developing a tracking system of individual (non-raining) clouds, their LWP and CDNC and the aerosol inside the cloud droplets over the cloud lifetime. Afterwards the clouds would need to be categorized by LWP to be able to analyse the growth of the in-cloud aerosol particles. This is beyond the scope of this study.

It is however known that the in-cloud aerosol size increases by processing in clouds. A reference for this was added.

*A possible mechanism to explain the negative LWP susceptibilities is the growth of aerosol particles in cloud droplets (by collisions of the cloud droplets with interstitial aerosol particles and heterogeneous chemistry; Hoose et al. 2008a) and release of the larger aerosol particles when the cloud droplets evaporate (as AIdry decreases for larger particles).*

P11L14: I am not sure I understand the reasoning here (and this is an important point) as to why AODdry is a better proxy than AIdry? AODdry is less sensitive to aerosol size than AIdry, but aerosol activation is quite sensitive to aerosol size.

This statement was ill formulated and subsequently removed. See also our response to the comments of Referee #1 (P11L15-16 and P16L8-11).

P11L27: Although the meteorological regimes are a good way to look at this, the split by humidity regimes may also confound different cloud or aerosol types. Maps of these sensitivities might be useful (at the authors' discretion)

The occurrence frequency of the environmental regimes is shown in Fig. 1b and 1c. One can see that there is a tendency for moist and dry as well as stable and unstable regimes to occur in different geographical regions although there is also some overlap of the regimes. This split of the regimes may confound different cloud types and it is also an intention of computing the susceptibilities for the different environmental regimes to assess susceptibilities for different cloud types (implicitly). Note however that non-raining and raining regimes occur in similar geographical regions and should therefore confound similar cloud and aerosol types. We focus in our study on the comparison between non-raining and raining regimes. Below are maps of the LWP susceptibility to $AI_{dry}$ from ECHAM6-HAM2 (E6_Ref) for the different environmental regimes. Note that the values shown in Fig. 4 are weighted averages of the susceptibilities on the maps below. The averaging is done over global oceans weighted by the occurrence frequency of aerosol-cloud data pairs.

*To assess the impact of environmental regimes, susceptibilities averaged over all grid boxes of each environmental regime (cf. Fig. 1b,c) are examined in this section.*

**E6_Ref**

$$(d \ln LWP)/(d \ln AI_{dry})$$

**ECHAM6-HAM2(dry)**

[Figure]

P11L31: The AI-CDNC relationship is mainly looking at aerosol activation - does wet scavenging really affect this, or is the change in the relationship in precipitating scenes indicative of differing aerosol types/cloud updraughts?

The analysis is done for low warm clouds only (cloud top pressures > 500 hPa and cloud top temperatures > 273.15 K). Furthermore the non-raining and raining regimes occur in similar geographical regions (cf. Fig. 1b,c). Non-raining and raining regimes should therefore confound similar cloud and aerosol types although the cloud updraft velocities may be different. The updraft velocities may be higher in the raining than in the non-raining scenes. This was added to the text.

*Part of the differences between raining and non-raining scenes may be due to different updraft velocities though, which may be higher in the raining than in the non-raining scenes.*

P12L2: based on Fig. 4a, I would have said that the regime variability in ECHAM using AIdry is similar, or even larger than the satellite products.

This was also not well formulated. The main point here was that in the satellite data the sign of the susceptibility changes between non-raining and raining regimes whereas in ECHAM6-HAM2 it is always positive. This has been reformulated. See also our response to your first comment (P1L23).

*When AIdry is used instead the magnitude of the LWP susceptibility is close to that of AATSR-CAPA and MODIS-CERES and the variability between environmental regimes in ECHAM6-HAM2 is similar to AATSR-CAPA. In most regimes, the LWP susceptibility to changes in AI or AIdry is larger in the non-raining than in the raining scenes and even negative in some regimes in the raining scenes for AATSR-CAPA, similar to the CDNC susceptibility.*

and:

*A reason that the effect of entrainment seems not to appear in the non-raining scenes in ECHAM6-HAM2 could be that cloud-top entrainment is not well represented in the model. ... At the coarse vertical resolution of a global climate model numerical artefacts like numerical entrainment (Lenderink and Holtslag, 2000) occur and the cloud top cooling that drives the turbulence in the boundary layer cannot be computed accurately (Stevens et al., 1999).*

P12L19: Is there a way of checking if sampling is the issue here? Are there some situations where MODIS/AATSR refuse to retrieve cloud/aerosol properties?

Sampling is generally not an issue of retrieval failure. The differences are related to:
1) AATSR samples along the full width of a 512 km swath whereas the MODIS-CERES data is along the CloudSat nadir view track
2) AATSR regional regressions are computed using four individual seasons and then averaged together to form the annual mean, while for MODIS-CERES, with its limited samples, regressions are computed using all 3.5 years (2006 - 2010) of data. Using this approach gives similar values to Lebsock et al. (2008), JGR who split this data into seasons (with worse data coverage).

*A reason could be the different sampling between AATSR-CAPA and MODIS-CERES where AATSR has a longer time series and wider swath. The MODIS-CERES data is along the CloudSat nadir view track.*

P12L30: Does alpha not depend on the cloud properties to some extent (if not these retrieved ones), when computing the fluxes from CERES broad-band radiances? Perhaps this is not a significant issue?

Alpha depends on the surface reflectance, cloud properties (cloud optical thickness and cloud effective radius), and solar zenith angle. It can be obtained by measuring the incoming and outgoing fluxes using CERES or derived from the cloud optical properties retrieved from MODIS. The advantage to CERES observations is that no assumptions are needed regarding the surface or cloud characteristics but the downside to this instrument is the coarser spatial resolution (20 km) compared to MODIS (1 km). The CERES observations are therefore well suited for intrinsic/extrinsic forcing calculations because the only key variables required are the fluxes and cloud fraction. Regarding, MODIS-CAPA, the cloud albedo is computed using BUGSrad and is accurate to within 5% of CERES (Christensen, M. W., Poulsen, C., McGarragh, G., and Grainger, R. G.: Algorithm Theoretical Basis Document (ATBD) of the Community Code for CLimate (CC4CL) Broadband Radiative Flux Retrieval (CC4CL-TOAFLUX) module, ESA Cloud CCI, 1, http://www.esa-cloud-cci.org, 2016b.).

P13L25: Fig. 7a shows drizzle water path, rather than LWP
Fig. 7b should have been referenced, this was corrected.
P13L29: This is not true for all relationships (e.g. Gryspeerdt et al., 2017). This might just mean that the AI-LWP relationship is not a good proxy for the strength of the aerosol influence on LWP.
We agree that this is not true for all relationships but Ghan et al. (2016) showed that it is true for the LWP susceptibility. Therefore, we specified in the text that this is true for several susceptibilities such as the LWP susceptibility and also mention that co-varying variables might affect the LWP susceptibility as well.

*Carslaw et al. (2013) and Ghan et al. (2016) found that present day variability is a poor proxy for the change due to anthropogenic aerosol for several susceptibilities such as the LWP susceptibility. Our results are similar to their findings as the difference between the prognostic and the diagnostic precipitation scheme leads to a weaker LWP response to anthropogenic aerosols (Sant et al., 2015) but a stronger LWP response determined by present day variability (Fig. 6). Note that co-varying variables might affect the LWP susceptibility as well.*

P14L16: Could these regions be drawn on the maps (perhaps in fig 1)
The regions were added to the revised Fig. 1a.

[Figure]

P15L1: Could these ERFaci values be compared with values determined from the model (PD-PI simulations)?

The ERF$_{aci}$ values for low warm clouds only (cloud top pressures > 500 hPa and cloud top temperatures > 273.15 K) were diagnosed from simulations with present day and pre-industrial aerosol emissions and added to the results. See our response to your comment P7L9.

P16L20: See earlier comment about model vs. satellite variability (P12L3)

This was also reformulated to point out that the change in sign of the LWP susceptibility only occurs for MODIS-CERES in the non-raining regimes and not for AATSR-CAPA or ECHAM6-HAM2.

*A differentiation of susceptibilities by different environmental regimes (precipitation, stability in the lower troposphere, RH in the lower free troposphere) revealed that AATSR-CAPA, MODIS-CERES and ECHAM6-HAM2 not always agree in their dependence on environmental regimes. The susceptibility of liquid water path is negative in non-raining scenes for MODIS-CERES but positive for AATSR-CAPA (and ECHAM6-HAM2). A negative LWP susceptibility in non-raining scenes has been interpreted as cloud top entrainment (Chen et al., 2014). Feedback processes such as cloud top entrainment that are missing or not well represented in ECHAM6-HAM2 are therefore not well constrained by the satellite observations. Further research with multiple satellite aerosol and cloud products could help to better understand such feedback processes and provide better constrains for climate models.*

References
Gryspeerdt, E. et al. (2017), Constraining the instantaneous aerosol influence on cloud albedo, Proc. Natl. Acad. Sci. USA, 114(19), 4899–4904, doi:10.1073/pnas.1617765114.

Khairoutdinov, M., and Y. Kogan (2000), A New Cloud Physics Parameterization in a Large-Eddy Simulation Model of Marine Stratocumulus, Mon. Wea. Rev., 128, 229, doi:10.1175/1520-0493(2000)128(0229:ANCPPI)2.0.CO;2.

Lebsock, M. D., G. L. Stephens, and C. Kummerow (2008), Multisensor satellite observations of aerosol effects on warm clouds, J. Geophys. Res., 113, D15205, doi:10.1029/2008JD009876.

Nakajima, T., Higurashi, A., Kawamoto, K. and Penner, J. E.: A possible correlation between satellite-derived cloud and aerosol microphysical parameters, Geophys. Res. Lett., 28(7), 1171–1174, doi:10.1029/2000GL012186, 2001.

Stier, P. (2016), Limitations of passive remote sensing to constrain global cloud condensation nuclei, Atmos. Chem. Phys., 16(10), 6595–6607, doi:10.5194/acp-16-6595-2016.

Suzuki, K., Stephens, G., van den Heever, S., and Nakajima, T.: Diagnosis of the warm rain process in cloud-resolving models using joint CloudSat and MODIS observations. J. Atmos. Sci., 68, 2655–2670, 2011.

Suzuki, K., Golaz, J.-C., and Stephens, G. L.: Evaluating cloud tuning in a climate model with satellite observations, Geophys. Res. Lett., 40, 4464–4468, doi:10.1002/grl.50874, 2013.

---

## Author Response (AR2)

**Response to editor review on *Unveiling aerosol-cloud interactions Part 2: Minimizing the effects of aerosol swelling and wet scavenging in ECHAM6-HAM2 for comparison to satellite data*, Atmos. Chem. Phys. Discuss., https://doi.org/10.5194/acp-2017-449**

David Neubauer, Matthew W. Christensen, Caroline Poulsen, Ulrike Lohmann

We would like to thank the editor for his careful handling of the manuscript.

The original comments are in black. Responses are in blue. *Modifications to the text are in green and italics*.

**Editor review:**

I have one concern: In almost all of your map plots there are wave structures. I dont think this is physical nor a model feature. I suspect some plotting artefact or model diagnostic artefact? Please check and possibly correct,

We have checked the diagnostics, the plotting scripts and the data. The wave structures is due to spurious numerical oscillations (SNOs), also known as Gibbs oscillations. Geil and Zeng (2015) analyzed 48 CMIP5 models and found that 48% of nonspectral models and 95% of spectral models show SNOs, which amplitude can be as large or greater than interannual variability. The wavelength of the SNOs depends on the resolution for spectral models (cf. Fig. 1 in Geil and Zeng, 2015). ECHAM6.1-HAM2.2 is a spectral model and is based on ECHAM6.1 (the main differences are the cloud and aerosol schemes). ECHAM6.1 is the atmospheric component of MPI-ESM analyzed by Geil and Zeng (2015). In their Fig. 1 they show the surface pressure field for MPI-ESM-MR (T63), which has the same spectral resolution (T63) as the ECHAM-HAM simulations in our study. In the surface pressure field for MPI-ESM-MR (T63) SNOs are visible.

We find that the visibility of the SNOs in the ECHAM-HAM simulations shows a weak dependence on the output frequency. Fig. X shows for example the mean LWP for 1995-2012 of the ECHAM6-HAM2 E6_Ref simulation. In Fig. Xa,c the SNOs computed from 3-hourly instantaneous output are better visible than the SNOs in Fig. Xb,d computed as a mean over all time steps. It could be that the SNOs depend on the wind field which may itself depend on the local time of day. When the model output is averaged over all time steps (Fig. Xb,d) the pattern of the SNOs could therefore become smaller whereas for the 3-hourly instantaneous output it would remain visible.

As the wave structures in our map plots are due to these SNOs we have added a statement for this at the end of 4.1.1:

*Note that the wave structures visible in Fig. 2 and some other figures are due to spurious numerical oscillations, which commonly appear in spectral models (Geil and Zeng, 2015).*

[Figure]

**Figure X**: Mean liquid water path (LWP) over the time period 1995-2012 for E6_Ref, a) computed from 3-hourly instantaneous values, b) accumulated over all model time steps, c) same as in a) with different color map, d) same as in b) with different color map.

References

[revised manuscript text omitted]

---

## Author Response (AR3)

Response to editor review on *Unveiling aerosol-cloud interactions Part 2: Minimizing the effects of aerosol swelling and wet scavenging in ECHAM6-HAM2 for comparison to satellite data*, Atmos. Chem. Phys. Discuss., https://doi.org/10.5194/acp-2017-449

David Neubauer, Matthew W. Christensen, Caroline A. Poulsen, Ulrike Lohmann

We would like to thank the editor for his careful handling of the manuscript.

The original comments are in black. Responses are in blue. *Modifications to the text are in green and italics*.

**Editor comment:**

thanks for the final comment added. It could be a bit longer, and please add the reference in the final paper submitted.

Our results do not seem to be affected (at least qualitatively) by spurious numerical oscillations (SNOs) as the SNOs of the susceptibilities (cf. Fig. 2) are weaker than in the cloud and aerosol input fields. Furthermore 
[revised manuscript text omitted]